# MEMORIZING LONG-TAIL DATA CAN HELP GENERALIZATION THROUGH COMPOSITION

**Mo Zhou**[*1]**, Haoyang Ma**[*2]**, Rong Ge** [2]
[1] University of Washington, [2] Duke University
`mozhou17@cs.washington.edu, haoyang.ma@duke.edu, rongge@cs.duke.edu`

## ABSTRACT

Deep learning has led researchers to rethink the relationship between memorization and generalization. In many settings, memorization does not hurt generalization due to implicit regularization and may help by memorizing long-tailed examples. In this paper, we consider the synergy between memorization and simple composition — the ability to make correct prediction on a combination of long-tailed features. Theoretically, we show that for a linear setting, memorization together with composition can help the model make correct predictions on rare test examples that require a combination of long-tailed features, even if such combinations were never observed in the training data. Experiments on neural network architecture on simple data show that the theoretical insight extends beyond the linear setting, and we further observe that the composition capability of the model depends on its architecture. [1]

## 1 INTRODUCTION

The relationship between memorization and generalization has always been an intriguing topic in deep learning. It has long been known that neural networks used for supervised learning can memorize noisy or even random labels (Zhang et al., 2017), and recent large language models can still memorize long text despite not being overparametrized (Carlini et al., 2019; 2022). Conventional wisdom from statistical learning theory suggests that memorization might be detrimental to generalization performance, yet neural networks often generalize well with memorization, sometimes even better compared to networks that do not memorize.

Trying to understand this relationship from a theoretical perspective has led to the idea of implicit regularization and benign overfitting (Belkin et al., 2019; Bartlett et al., 2020; Belkin et al., 2020; Hastie et al., 2022; Gunasekar et al., 2017), where the training process and architecture choices of neural networks prefer certain solutions that have good generalization despite memorizing the training data. Another interesting line of work (Feldman, 2020; Feldman & Zhang, 2020) demonstrated that memorization can actively help generalization by capturing long-tail behaviors in the training data. If the test data contain examples that are similar to training examples, it is more likely for the neural network to make correct predictions.

However, the similarity between the test and training data alone cannot explain all the benefits of memorization. Recent large models exhibit several surprising properties including *composition* – the ability to produce correct results by combining two or more pieces of information learned during training. Composition has received a lot of attention (Hupkes et al., 2020; Baroni, 2020). Composition allows models to perform tasks that may not appear in the training data, resulting in stronger notions of out-of-distribution (OOD) generalization. Intuitively, memorizing long-tail examples and composition are synergistic: a model that is capable of composition may leverage two or more memorized long-tail examples to create/predict something that is highly unlikely in the original training data. In this paper, we try to demonstrate this behavior on simplistic models.

Our theoretical approach focuses on a linear setting where features appear in different frequencies, and each sample consists of a sparse combination of features. As the model is linear, it automatically

---

[*]Equal contribution.
[1]Code available at `https://github.com/mhy-666/long-tail-memorization-composition`.

achieves simple composition — if individual features are learned correctly, their combinations can also be predicted correctly. We give guarantees for out-of-distribution generalization where the test data contain examples that may be combinations of multiple infrequent features and have very low probability of appearing in the training set.

To complement our theoretical results, we construct simple examples that require simple composition — prediction of the sum of digits for 3-digit numbers. Our experiment shows that in such cases one can indeed train simple networks that are capable of composition and memorization helps with both in-distribution and out-of-distribution (OOD) generalization. We also observe that the model architecture plays an important role in determining whether the model has composition capabilities.

## 1.1 OUR RESULTS

Informally, our theoretical model considers a data distribution $\mathcal{D}$ over vectors $\boldsymbol{x}$, where each feature $x_i$ appears independently with probability $p_i$. These probabilities decay as $p_1 \geq p_2 \geq \ldots \geq p_d$ (for example, $p_i \sim i^{-\alpha}$ follows a power-law decay), and each data is a linear combination of $s$ features. As a result, features with larger $p_i$ tend to appear frequently in the data, while those with smaller $p_i$ may appear only a few times or not at all. We refer to the former as common features and the latter as long-tail features.

Our goal is to learn a target model of the form $f_*(\boldsymbol{x}) = \langle \boldsymbol{\beta}_*, \boldsymbol{x} \rangle$ using $n$ training samples of the form $y = f_*(\boldsymbol{x}) + \xi$, where the label noise $\xi \sim \mathcal{N}(0, \sigma^2)$ and the noise level $\sigma \geq 0$. We focus on the minimum $\ell_2$-norm solution, which memorizes the training set in our overparametrized setting.

We show that under this setup, the model can not only reliably learn common features that appear at least $\tilde{\Omega}(1)$ times[2], but also recover certain long-tail features that appear only once. This enables the model to perform well when tested on combinations of such features that were never seen together during training:

**Theorem 1** (Informal). *Let $\widetilde{\mathcal{D}}$ be a test distribution that differs from the training distribution $\mathcal{D}$ only in that each sample from $\widetilde{\mathcal{D}}$ contains a composition of $t$ long-tail features, each of which appeared exactly once in the training data. Then, for the minimum $\ell_2$-norm solution the test loss (expected loss on $\mathcal{D}$) remains small, and the out-of-distribution loss (expected loss on $\widetilde{\mathcal{D}}$) is at most $\sigma^2 t$.*

The main challenge is that the frequency $p_i$ of common features can be as low as $\tilde{\Theta}\left(\frac{1}{n}\right)$, resulting in a high variance. This issue is even worse for long-tail features, which appear even less frequently. As a result, standard concentration bounds are insufficient in this setting. To address this, we leverage the combinatorial structure of the data matrix $\boldsymbol{X}$ and show that, under a suitable decay of $p_i$, most data points contain at most one long-tail feature. This structural insight enables us to establish meaningful learning guarantees, even when some features are extremely rare and variances are large.

Empirically, we first verify the theoretical analysis in the linear setting. We then show that similar phenomena happen for simple neural networks. More concretely, we consider a "heavy-tailed" version of MNIST dataset where digits appear in decreasing frequencies, with 9 being the rarest. To introduce compositionality, each input consists of 3 randomly sampled MNIST images, and the task is to predict the sum of the 3 digits.

We evaluate the composition capability of model through its OOD performance on test examples that contains at least one 9, with the other two numbers sampled uniformly. Most of the test cases involve 3-digit combinations that never appear during training. We observe that ResNet models exhibit compositionality when they process each digit individually to extract features, and then aggregated these features by a simple network — even if the network has not seen a particular 3-digit combination, the prediction is still accurate if it has seen each digit enough number of times. In contrast, a ResNet that processes the 3 images together as a single input performs significantly worse.

To further investigate memorization effect on generalization, we insert images from Omniglot in the training data, where each Omniglot class only appear *exactly once*. Experiments show that models with compositional ability can correctly predict on test data that involves two Omniglot images, provided that the model memorizes training data. However, models that either do not memorize due to a large weight decay or lack compositional ability perform significantly worse.

---

[2]We use $\tilde{O}, \tilde{\Omega}, \tilde{\Theta}$ to hide polylog factors.

## 1.2 RELATED WORKS

**Memorization and long-tail data**    Real-world data often follow heavy-tailed distributions, where rare examples capture unique sub-concepts (Zhu et al., 2014). The works most related to ours are Feldman (2020) and Feldman & Zhang (2020). These papers demonstrated that memorization is necessary to achieve near-optimal generalization on long-tail data. Our work builds on this insight and takes it further, suggesting that memorization can also support the composition of long-tail data and features.

**Memorization in learning**    Researchers have discovered that deep neural networks can fit (i.e., "memorize") arbitrary labels and still generalize, challenging the classical bias–variance intuition (Zhang et al., 2017). Since then, there has been growing interest in understanding memorization from various perspectives (see the survey by Wei et al. (2024) for a comprehensive overview).

Various forms of memorization have been explored in different contexts, including label memorization in supervised learning (Feldman, 2020; Feldman & Zhang, 2020), counterfactual memorization (Zhang et al., 2023), and exact memorization in language models (Tirumala et al., 2022). These findings have raised privacy concerns about the unintended memorization of sensitive data (Carlini et al., 2019). In response, forgetting mechanisms and unlearning algorithms have been developed to remove such content without full retraining (e.g., Cao & Yang (2015); Garg et al. (2020)). In contrast, our work focuses on the positive side of memorization, similar to Feldman (2020), showing how it can support compositional learning over long-tail data distributions.

**Compositional Generalization**    Generalizing to unseen compositions of known data or features has a long history (Fodor & Pylyshyn, 1988) and remains a significant challenge. In recent years, benchmarks such as SCAN (Lake & Baroni, 2018) and CoGS (Kim & Linzen, 2020) have been introduced to evaluate models' compositional abilities. Extensive analyses using these benchmarks have revealed that some language and vision-language models struggle with aspects like word order sensitivity, relationship recognition, or counting (Hessel & Schofield, 2021; Parcalabescu et al., 2021; Yuksekgonul et al., 2022). Nonetheless, recent work suggests that current language models do exhibit some capacity for compositional skill (Bubeck et al., 2023; Zhao et al., 2025).

To improve compositional generalization, researchers have proposed various methods, including data augmentation (Andreas, 2019), architectural designs (Andreas et al., 2016; Hudson & Manning, 2018), and meta-learning approaches (Conklin et al., 2021). Efforts to better understand compositional generalization continue in recent years (Wiedemer et al., 2023; Tang et al., 2025; Arora & Goyal, 2023; Lippl & Stachenfeld, 2024). Our work builds on this line of inquiry, but we focus more on the connection between memorization and compositionality.

## 2 PRELIMINARIES

In this section, we set up the details of our theoretical model.

**Notation**    Let $[n] = 1, 2, \ldots, n$. Vectors and matrices are denoted in bold. For a vector $\boldsymbol{\beta} \in \mathbb{R}^d$ and a subset $A \subseteq [d]$, define $\boldsymbol{\beta}_A := \sum_{i \in A} \beta_i \boldsymbol{e}_i$, keeping entries in $A$ and setting others to 0, where $\boldsymbol{e}_i$ is the standard basis. We use standard notation $O, \Omega, \Theta, \lesssim, \gtrsim$ to hide constants, and $\widetilde{O}, \widetilde{\Omega}, \widetilde{\Theta}$ to also hide logarithmic factors like $\log(d)$ and $\log(n)$.

**Target Model**    We aim to learn the following linear model:

$$f_*(\boldsymbol{x}) = \langle \boldsymbol{\beta}^*, \boldsymbol{x} \rangle.$$

Here we assume largest coefficient is bounded $\max_i |\beta_i| = \Theta(1)$.

**Data**    Our data model is designed to capture the common long-tail distribution of features in real-world data (Zhu et al., 2014). We assume each data point has only a few active features, with feature frequencies varying according to a long-tail pattern. For instance, in image classification each image may contain only a few objects, while the overall feature set is large and follows a long-tail distribution.

Formally, the data vectors $\boldsymbol{x} = (x_1, \ldots, x_d)^\top$ are drawn from the distribution $\mathcal{D}(\{p_i\})$ defined as follows: given probability $1 \geq p_1 \geq \ldots \geq p_d > 0$, each feature $x_i \in \{0, 1, -1\}$ is independently

sampled from a Bernoulli distribution with parameter $p_i$, followed by a random sign flip:

$$\mathbb{P}(x_i = 0) = 1 - p_i, \quad \mathbb{P}(x_i = 1) = \mathbb{P}(x_i = -1) = \frac{p_i}{2}$$

Let $\boldsymbol{\Sigma} = \mathbb{E}[\boldsymbol{x}\boldsymbol{x}^\top] = \mathrm{diag}(p_1, \ldots, p_d)$ be the covariance matrix, and we write $\mathcal{D}(\{p_i\})$ as $\mathcal{D}(\boldsymbol{\Sigma})$.

We generate $n$ data points $\{(\boldsymbol{x}_i, y_i)\}_{i=1}^n$ i.i.d. according to this process with label noise $\xi$:

$$y_i = f_*(\boldsymbol{x}_i) + \xi_i, \quad \text{where } \boldsymbol{x}_i \sim \mathcal{D}(\boldsymbol{\Sigma}), \ \xi_i \sim \mathcal{N}(0, \sigma^2 \boldsymbol{I}). \tag{1}$$

In many cases, we assume that the feature probabilities $p_i$ follow a power-law decay:

$$p_i := \min\{1, s \cdot i^{-\alpha} / Z_\alpha\} \tag{2}$$

for some constant $\alpha > 0$ with normalization constant $Z_\alpha = \sum_{i \leq d} i^{-\alpha}$. The truncation at 1 is only to make sure $p_i \in [0, 1]$ as a valid probability. Note that $\mathbb{E}[\|\boldsymbol{x}\|_2^2] \approx s$ (ignoring the truncation) so each data $\boldsymbol{x}$ will be roughly $s$-sparse. In the rest of the paper, for a data $\boldsymbol{x}$ we will often call the top-$k$ entries $\boldsymbol{x}_{\leq k}$ as common features and remain entries $\boldsymbol{x}_{>k}$ as (long-)tail features for a threshold $k$ chosen later. Also denote $p_{>k} := \sum_{i>k} p_i$ as the tail probability and $p_{\leq k} := \sum_{i \leq k} p_i$ as the head probability. The tail probability can be bounded by the claim below.

**Claim 1.** *For $p_i$ follows power law decay* (2) *with $\alpha = 1 + c_\alpha$ for any constant $c_\alpha > 0$ and $p_k < 1$ (i.e., $k \geq \Theta(s^{1/\alpha})$), we have $p_{>k} = \Theta(sk^{1-\alpha})$ and $p_k = \Theta(sk^{-\alpha})$.*

Our data model is intended to reflect the long-tail structure, and the power-law decay assumption is common in prior theoretical analyses (e.g., (Bartlett et al., 2020; Hastie et al., 2022; Lin et al., 2024)).

**Choice of parameters** Let $c_{\log} = \mathrm{polylog}(d)$ be a large enough polylog factor. We assume $s < c_{\log} < k < n/c_{\log} < d/c_{\log}^2$ and $n^c > s + \ln d$ for any constant $c > 0$. A concrete example is $s = \Theta(1)$, $n = d^c$ for constant $c \in (0, 0.99)$ and $c_{\log} < k < n/c_{\log}$. Casual readers may assume $s \ll \mathrm{polylog}(d) \ll k \ll n \ll d$. The threshold $k$ between common and long-tail features will be chosen later. Intuitively we require $p_k = \tilde{\Theta}(1)$ so common features appear at least polylog times.

**Algorithm and loss** We use a linear model and run gradient descent on square loss. That is

$$L(\boldsymbol{\beta}) = \frac{1}{n} \sum_{i=1}^n (\langle \boldsymbol{\beta}, \boldsymbol{x}_i \rangle - y_i)^2 = \frac{1}{n} \|\boldsymbol{X}\boldsymbol{\beta} - \boldsymbol{y}\|_2^2.$$

We run gradient descent from 0 initialization, so the final estimator is

$$\hat{\boldsymbol{\beta}} = (\boldsymbol{X}^\top \boldsymbol{X})^\dagger \boldsymbol{X}^\top \boldsymbol{y}.$$

This estimator is also known as the min-$\ell_2$-norm interpolator/solution. Similar overparametrized linear models and kernel-models are standard in previous theoretical works (e.g., Bartlett et al. (2020); Belkin et al. (2019); Jacot et al. (2018)).

We focus on the overparametrized regime where $d \gg n$. In the noiseless case $\sigma = 0$, such interpolator memorizes all training data. In noisy case $\sigma > 0$, the minimum training loss may be above 0 even in such overparametrized setting (because there may be many samples that only use a smaller number of features), so the solution may not interpolate the data. Still, it represents a global minimum with the smallest $\ell_2$ norm among all such minima.

Since both our data generation process and predictor are linear, composition happens automatically — if a model learns the $\beta$ values for a subset of features accurately, it will perform well for any data that uses a combination of features from this subset.

**Performance Measurements: In- and Out-of-Distribution Test Loss** For in-distribution performance, we use the standard test loss:

$$\mathbb{E}_{x \sim \mathcal{D}}[(\hat{\boldsymbol{\beta}}^\top \boldsymbol{x} - \boldsymbol{\beta}^{*\top} \boldsymbol{x})^2].$$

To evaluate the model's composition capability (its ability to perform well on rare test examples that are combinations of training features never appear in training data), we measure its OOD performance using the test loss defined below.

Let $\mathcal{F} = \{j \in [d] : \exists i \in [n] \; s.t. \; (\boldsymbol{x}_i)_j \neq 0\}$ be the set of features that appear in the training data. Let $\widetilde{\mathcal{F}} \subseteq \mathcal{F}$ be a subset of these features. We define $\mathcal{D}_{\widetilde{\mathcal{F}}}$ as a modified distribution of $\mathcal{D}$, where each feature $x_i$ in $\widetilde{\mathcal{F}}$ is guaranteed to be non-zero in every sample, while the rest of the data is generated according to the original distribution $\mathcal{D}$: each feature $x_i$ not in $\widetilde{\mathcal{F}}$ appears with its original frequency $p_i$, and non-zero values are sampled uniformly from $\pm 1$.

For example, $\mathcal{D}_{\{2k,4k\}}$ denotes a distribution where $x_{2k}, x_{4k} \neq 0$. While the model might encounter training examples where only $x_{2k}$ or $x_{4k}$ is non-zero, it is very likely that it does not see examples where both are non-zero simultaneously. The OOD test loss is defined as:

$$\mathbb{E}_{x \sim \mathcal{D}_{\widetilde{\mathcal{F}}}}[(\hat{\boldsymbol{\beta}}^\top \boldsymbol{x} - \boldsymbol{\beta}^{*\top} \boldsymbol{x})^2].$$

# 3 MEMORIZATION HELPS OUT-OF-DISTRIBUTION GENERALIZATION

In this section, we present our main results, which show that memorization can help with generalization by composition of long-tail features, as measured by test loss and OOD loss. We present results for both the noiseless case ($\sigma = 0$) and the noisy case ($\sigma > 0$), respectively.

## 3.1 NOISELESS CASE

We start with noiseless case. We show below the test performance of the estimator $\hat{\boldsymbol{\beta}}$ that goes to 0 when $n$ goes large. The intuition behind the proof is that we can recover all feature that shows up at least $\Theta(\ln^2 d)$ times, so their estimation error goes to 0.

**Theorem 2** (Test loss). *Suppose data generated from* (1) *with noise* $\sigma = 0$*, then with probability at least 0.99 over the data generation process, the following hold with $k$ satisfying $np_k = \Theta(\ln^2 d)$ and $p_{>k} \leq 1 - c_p$ with any constant $c_p$:*

$$\mathbb{E}_{x \sim \mathcal{D}}[(\hat{\boldsymbol{\beta}}^\top \boldsymbol{x} - \boldsymbol{\beta}^{*\top} \boldsymbol{x})^2] \lesssim p_{>k} + \frac{k \ln^4 d}{n^2 p_{>k}} + \frac{\ln^4 d}{n} + \frac{k p_{>k}^2 \ln^4 d}{n} + p_{>k}^3 \ln^4 d.$$

*When data follows power law decay* (2) *with $\alpha = 1 + c_\alpha$ with any constant $c_\alpha > 0$*

$$\mathbb{E}_{x \sim \mathcal{D}}[(\hat{\boldsymbol{\beta}}^\top \boldsymbol{x} - \boldsymbol{\beta}^{*\top} \boldsymbol{x})^2] \lesssim s \left(\frac{\ln^2 d}{ns}\right)^{1 - \frac{1}{\alpha}}$$

Moreover, we establish the following OOD generalization guarantee, which reflects the model's compositional ability and shows that it can still perform well despite memorizing all training data. As shown in the proof, this is made possible by the composition of features that the model memorizes.

**Theorem 3** (OOD performance). *Under same condition as Theorem 2, then with probability at least 0.99 over the data generation process, there is a feature set $\hat{\mathcal{F}} \subseteq \mathcal{F}$ with size $|\hat{\mathcal{F}}| \geq (1 - \Theta(\max\{p_k/p_{>k}, p_{>k}^2 \ln^2 d\}))|\mathcal{F}|$ that contains almost all features show up in data such that for any feature set $\widetilde{\mathcal{F}} \subseteq \hat{\mathcal{F}}$ we have*

$$\mathbb{E}_{x \sim \mathcal{D}_{\widetilde{\mathcal{F}}}}[(\hat{\boldsymbol{\beta}}^\top \boldsymbol{x} - \boldsymbol{\beta}^{*\top} \boldsymbol{x})^2] \lesssim p_{>k} + \frac{k \ln^4 d}{n^2 p_{>k}} + \frac{\ln^4 d}{n} + \frac{k p_{>k}^2 \ln^4 d}{n} + p_{>k}^3 \ln^4 d.$$

*When data follows power law decay* (2) *with $\alpha = 1 + c_\alpha$ with any constant $c_\alpha > 0$ we have* $|\hat{\mathcal{F}}| \geq (1 - \Theta\left(\max\left\{\left(\frac{\ln^2 d}{ns}\right)^{\frac{1}{\alpha}}, \frac{s^{\frac{2}{\alpha}}(\ln d)^{6 - \frac{4}{\alpha}}}{n^{2 - \frac{2}{\alpha}}}\right\}\right))|\mathcal{F}|$ *and*

$$\mathbb{E}_{x \sim \mathcal{D}_{\widetilde{\mathcal{F}}}}[(\hat{\boldsymbol{\beta}}^\top \boldsymbol{x} - \boldsymbol{\beta}^{*\top} \boldsymbol{x})^2] \lesssim s \left(\frac{\ln^2 d}{ns}\right)^{1 - \frac{1}{\alpha}}.$$

The proof ideas for both results are similar. We show that all features that appear at least $\Theta(\ln^2 d)$ times can be recovered, as well as a significant fraction of long-tail features that appear less frequently in the training data. Intuitively, this is the best we can hope for, since common features may appear only $\omega(1)$ times, and long-tail features may appear just once. We provide more details in Section 4.

These results suggest that memorization can be useful for learning compositions involving long-tail features. This may offer insight into why memorization is necessary in certain learning problems that require compositional ability. We further validate our findings in the experimental results (Section 5).

## 3.2 Noisy case

We now extend the above noiseless case ($\sigma = 0$) to the noisy case ($\sigma > 0$). Similarly, we show the test loss is small (roughly $\Theta(\sigma^2 k/n)$).

**Theorem 4** (Test loss). *Suppose data generated from* (1) *with noise* $\sigma \leq 1$*, with probability at least 0.99 over the data generation process, the following hold with $k$ satisfying $np_k = \Theta(\ln^2 d)$, $p_{>k} \leq 1 - c_p$ with any constant $c_p$ and $np_{>k}^2 < c$ for a small enough constant $c$:*

$$\mathbb{E}_{\boldsymbol{X}_{\leq k}}[\mathbb{E}_{x \sim \mathcal{D}}[(\hat{\boldsymbol{\beta}}^\top \boldsymbol{x} - \boldsymbol{\beta}^{*\top} \boldsymbol{x})^2]] \lesssim p_{>k} + \sigma^2 \left( \frac{k \ln d}{n} + \left( \frac{k^2 \ln^2 d}{n} + \ln d \right) p_{>k} \ln d \right)$$

*where the first expectation is over data generating process of common part of training data $\boldsymbol{X}_{\leq k}$.*

*When data follows power law decay* (2) *with $\alpha = 2 + c_\alpha$ with any constant $c_\alpha > 0$*

$$\mathbb{E}_{\boldsymbol{X}_{\leq k}}[\mathbb{E}_{x \sim \mathcal{D}}[(\hat{\boldsymbol{\beta}}^\top \boldsymbol{x} - \boldsymbol{\beta}^{*\top} \boldsymbol{x})^2]] \lesssim s \left( \frac{\ln^2 d}{ns} \right)^{1 - \frac{1}{\alpha}} + \sigma^2 \left( \frac{\ln^2 d}{ns} \right)^{1 - \frac{1}{\alpha}} s \ln^5 d$$

Similar to the noiseless case, we have the following OOD guarantee that measure model's compositional ability: the error is roughly $\sigma^2 t$, where $t$ is the number of long-tailed features present in the OOD distribution.

**Theorem 5** (OOD performance). *Under same condition as Theorem 4, then with probability at least 0.99 over the data generation process, consider any feature set $\widetilde{\mathcal{F}} \subseteq \mathcal{F} \setminus \mathcal{F}_0$ contains any long tail features show up in data, we have*

$$\mathbb{E}_{\boldsymbol{X}_{\leq k}}[\mathbb{E}_{x \sim \mathcal{D}_{\widetilde{\mathcal{F}}}}[(\hat{\boldsymbol{\beta}}^\top \boldsymbol{x} - \boldsymbol{\beta}^{*\top} \boldsymbol{x})^2]] \lesssim p_{>k} + \sigma^2 |\widetilde{\mathcal{F}}| \left( \frac{k^2 \ln^2 d}{n} + \ln d \right),$$

*where the first expectation is over data generating process of common part of training data $\boldsymbol{X}_{\leq k}$.*

*When data follows power law decay* (2) *with $\alpha = 2 + c_\alpha$ with any constant $c_\alpha > 0$*

$$\mathbb{E}_{\boldsymbol{X}_{\leq k}}[\mathbb{E}_{x \sim \mathcal{D}}[(\hat{\boldsymbol{\beta}}^\top \boldsymbol{x} - \boldsymbol{\beta}^{*\top} \boldsymbol{x})^2]] \lesssim s \left( \frac{\ln^2 d}{ns} \right)^{1 - \frac{1}{\alpha}} + \sigma^2 |\widetilde{\mathcal{F}}| \ln d.$$

These results suggest that the benefits of memorization for compositional learning persist even in the presence of label noise, although they require a stronger assumption on the frequency decay.

## 4 Proof idea for noiseless case

In this section, we present the proof ideas for the noiseless case. The proof for the noisy case follows a similar idea, but is slightly more complex.

A common strategy in analyzing the least squares estimator is to apply matrix concentration techniques to estimate $\boldsymbol{X}^\top \boldsymbol{X}$ and its inverse. However, in our setting, such a matrix is not going to concentrate for long-tail features that are expected to appear less than a constant number of times. Whether a feature appears or not has huge impact on the pseudoinverse $(\boldsymbol{X}^\top \boldsymbol{X})^\dagger$. Instead of concentration, we rely on the combinatorial structure of the data matrix $\boldsymbol{X}$, as described in the lemma below. Roughly speaking, the lemma shows that most data points contain either zero or one long-tail feature.

**Lemma 6** (Structure of $\boldsymbol{X}$, informal of Lemma A.1). *Suppose data generated from* (1)*, with probability at least 0.99, the following hold with $k$ satisfying $np_k = \Theta(\ln^2 d)$ and $p_{>k} \leq 1 - c_p$ with any constant $c_p$:*

1. *There exist at least $\Theta(n(1 - p_{>k}))$ samples whose non-zero entries are only within first $k$ entries. That is*

$$|S_0| := |\{i \in [n] : (\boldsymbol{x}_i)_j = 0, \forall j > k\}| = \Theta(n(1 - p_{>k})).$$

*Moreover, the dimension spanned by these samples in $S_0$ is $k$.*

2. *There exist at least $\Theta(np_{>k}(1 - p_{>k}))$ samples that only have exactly one non-zero entry in long-tail entries (not in the first $k$ entries). That is*

$$|S_1| := |\{i \in [n] : \exists \ell > k \text{ such that } (\boldsymbol{x}_i)_\ell \neq 0 \text{ and } (\boldsymbol{x}_i)_j = 0, \forall j > k, j \neq \ell\}| = \Theta(np_{>k}(1-p_{>k})).$$

3. *Denote the set of features show up in $S_0$ and $S_1$ as*

$$\mathcal{F}_0 = \{j \in [d] : \exists i \in S_0 \text{ s.t. } (\boldsymbol{x}_i)_j \neq 0\}, \quad \mathcal{F}_1 = \{j \in [d] : \exists i \in S_1 \text{ s.t. } (\boldsymbol{x}_i)_j \neq 0\}.$$

*We have the total number of features shown up in data is $|\mathcal{F}| = k + \Theta(np_{>k})$ and the number of features shown up in $S_0$ and $S_1$ is at least*

$$|\mathcal{F}_0 \cup \mathcal{F}_1| \geq \left(1 - \Theta\left(\max\{p_k/p_{>k}, p_{>k}^2 \ln^2 d\}\right)\right)|\mathcal{F}|$$

Note that in the noiseless case, we can achieve 0 training loss. Thus we know every sample must have 0 training loss. From the lemma above, we can see that there are $\Theta(n(1 - p_{>k}))$ samples involving only the $k$ common features. Therefore, the common component $\hat{\boldsymbol{\beta}}_{\leq k}$ is uniquely determined by these data points when $k < n$.

Moreover, once the common features $\hat{\boldsymbol{\beta}}_{\leq k}$ are identified, many long-tail features $\hat{\beta}_i$, those that appear only alongside common features and not with other long-tail features, can also be exactly recovered. This is formalized in the following lemma:

**Lemma 7** (Recovery of $\beta^*$). *Suppose data generated from (1) with noise $\sigma = 0$, for $k$ that satisfies the condition of Lemma 6, we have*

1. *Common features (Top-$k$ entries) are recovered: $\hat{\boldsymbol{\beta}}_{\leq k} = \boldsymbol{\beta}^*_{\leq k}$*

2. *Most of long tail features shown up in data X(last-$d - k$ entries) are recovered:*

$$\left|\left\{i \in [d] : \hat{\beta}_i = \beta^*_i\right\}\right| / |\mathcal{F}| \geq 1 - \Theta(\max\{p_k/p_{>k}, p_{>k}^2 \ln^2 d\}).$$

3. $\left\|\hat{\boldsymbol{\beta}}_{\mathcal{F}\setminus(\mathcal{F}_0\cup\mathcal{F}_1)}\right\|_2 \leq \left\|\boldsymbol{\beta}^*_{\mathcal{F}\setminus(\mathcal{F}_0\cup\mathcal{F}_1)}\right\|_2$ *and* $\hat{\beta}_i = 0$ *for* $i \notin \mathcal{F}$.

Using the parameter recovery result above, we can derive the performance guarantees stated in Theorem 2 and Theorem 3, by applying Claim 1.

In the noisy case, we require stronger tail probability decay assumption of data $X$ to show that (with high probability) every data point contains at most one long-tail feature. Even with this assumption, the key challenge for the noisy case is that the minimum $\ell_2$ norm solution may not have 0 training loss, hence we can no longer say that the training loss is 0 on every individual training samples as in the noiseless case. However, we show that even in this setting, if a long-tail feature only appear in one sample, then the training loss for that particular sample must be 0 and the long-tail feature will be estimated correctly (up to noise level $\sigma$) with high probability. Similar argument can be generalized when long-tail feature appear in few training samples. We leave the details to Appendix B.

## 5 EXPERIMENT RESULTS

In this section, we first validate our theoretical results empirically, and then we show that similar ideas can be extended to simple neural network models. All experiments, unless otherwise specified, reach near-zero training loss and therefore memorize the training data.

### 5.1 EXPERIMENTS ON LINEAR MODEL

We follow the setup in Section 2 where the parameters are chosen as: $n = 1000$, $d = 10000$, $s = 5$, $np_k = 10$ and $\boldsymbol{\beta}^* = (\beta^*_1, \ldots, \beta^*_d)^\top$ with $\beta^*_i = i^{-0.1}$, and report results averaging over 50 runs. In this setting the most frequent "long-tail" features would appear 10 times in expectation. As we can see in Figure 1b, for most values of $\alpha$ the error on the long-tail features is roughly proportional to the noise level $\sigma$. This is the same as the theoretical prediction as many of these long-tail examples appear only once and their error is slightly larger than the noise, while long-tail examples that appear

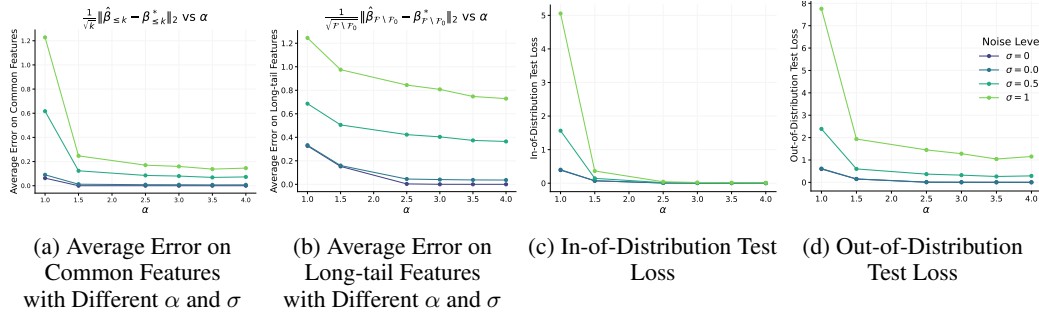

(a) Average Error on Common Features with Different $\alpha$ and $\sigma$

(b) Average Error on Long-tail Features with Different $\alpha$ and $\sigma$

(c) In-of-Distribution Test Loss

(d) Out-of-Distribution Test Loss

Figure 1: (a)(b) Average Error on Common and Long-tail Features on Linear Model; (c)(d) In- and Out-of-Distribution Test Loss on Linear Model

more often can have a smaller error. Figure 1a shows that common features are learned well when $\alpha$ is not too small. Figure 1c1d shows that the in-distribution and out-of-distribution test losses are small as in Theorem 2, 3, 4 and 5. Since the test losses for $\sigma = 0$ and $\sigma = 0.05$ are very small, their curves overlap in the plot.

We also observe that when $\alpha$ is small ($\alpha = 1$), both the in-distribution and out-of-distribution test losses become higher, and the feature errors also increase. This is expected because in such cases the long-tail features appear more often in the training data, not often enough for concentration bounds but often enough to weaken the combinatorial structure our analysis relies on.

## 5.2 TASK THAT REQUIRE SIMPLE COMPOSITION

Next, we consider a simple task that would benefit from composition. We constructed a new dataset based on MNIST. Each sample in this dataset consists of three randomly selected MNIST digit images in MNIST *training* set, stacked along the channel dimension. The task is to predict the sum of 3 digits. Specifically, given that original MNIST images are of size $28 \times 28$, our new samples take the form of $3 \times 28 \times 28$, where each "channel" corresponds to a different digit. The digit selection also follows Zipf's law distribution, meaning that lower digits (e.g., '0') appear more frequently, while higher digits (e.g., '9') appear less frequently.

To investigate the composition behavior and OOD generalization, every test sample is required to contain at least one digit '9', introducing a distribution shift between the training and test sets. All test samples are drawn from the MNIST *test* set. Importantly, this ensures that the digit '9's used at test time have never been seen by the model during training.

**Model Architecture**  We consider several models using ResNet-18 as our backbone. The first few models process each channel separately through a ResNet-18 network. That is, given a sample with three stacked digit images, we compute the feature representation $g_i$ (second to last layer) and label ($f_i$) for each channel independently using the standard ResNet-18 forward propagation. The final model output is obtained by some aggregation over the individual channel outputs. For **Sum**, the final output is simply the sum of $f_i$'s; for **Linear**, the final output is a trained linear layer on top of the second-to-last layer $g_i$'s (where the 3 features are concatenated); for **2-layer**, the final output is a trained 2-layer network on top of the second-to-last layer features $g_i$'s. Both the ResNet-18 part and the aggregation layers are trained simultaneously.

We also consider **Cross-channel**, which is a ResNet-18 model that is directly applied to the stacked images as if the channels are color channels (therefore the information between different images are mixed up starting from layer 1).

The training set has 32,000 samples, with number 0 appear 77,435 times and number 8 appear 955 times. We change the number of times 9 appears to study the relationship between that and the test performance. For the test set, we enforce that each sample has at least one position equal to number 9, where the two other positions are sampled uniformly from 0-4. Note that with the number of samples we have in the training data, with high probability most number combinations in the test set will not appear in the training set.

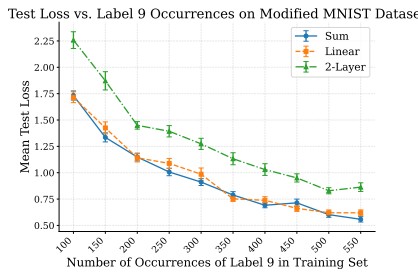 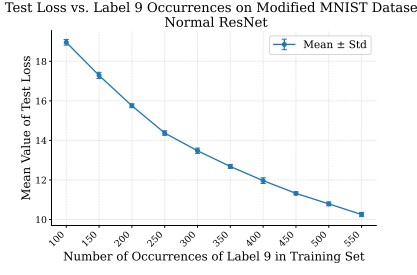

(a) Test Loss of Sum, Linear, and 2-Layer Models  (b) Test Loss of Cross-channel Model

Figure 2: Test Performance Comparison Across **Sum, Linear, 2-layer** and **Cross-channel** Models

Our results Figure 2a show that **Sum, Linear, 2-layer** models all perform well on the test distribution. Since the digit '9' is rare in the training data and therefore not learned well by the models, this good performance suggests that these models exhibit some form of compositional behavior by leveraging memorized instances of '9'. On the other hand, Figure 2b shows that **Cross-channel** model performs significantly worse on the test distribution, indicating that compositional capability still depends on the model architecture.

## 5.3 EFFECT OF MEMORIZATION

To understand the effect of memorization, we further introduce even rarer examples. We consider a similar setting as the previous subsection. Except that in the training data, we choose 10 classes from Omniglot (Lake et al., 2015) and select 1 image from each class. That image is assigned a label of 0-9. Each Omniglot image only appears exactly once in the training data (as one of the 3 images in one of the training data). The task is still to predict the sum of labels for the 3 images.

For the test data, we consider images which contain two of the images from Omniglot (for the images that have appeared in the training data). Such test examples never appear in the training data and would not be close to any of them in Euclidean or embedding distance.

As in Table 1, despite only having seen the data once and the test data being significantly different, for **Sum, Linear, 2-layer** models the performance on the test set is still reasonable when we don't use any regularization and make sure the model memorizes the training data. On the other hand, with weight decay, even though the training loss only increased slightly, the test performance become significantly worse. This suggests that memorization enables the model to develop a form of compositional ability that leverages the memorized Omniglot data, resulting in better generalization for memorizing models compared to those that do not memorize. **Cross-channel** model does not perform well in this setting, further supporting our findings that the composition ability depends on architecture.

Table 1: Memorization behavior of ResNet variants under different weight decay (WD) settings

| Model | WD = 0 (Memorization) | WD = 0.5 (No Memorization) |
|---|---|---|
| Sum | Test/Train Loss: 0.1760/0.0004 | Test/Train Loss: 2.8744/0.0245 |
| Linear | Test/Train Loss: 0.1662/0.0004 | Test/Train Loss: 1.5539/0.0505 |
| 2-layer | Test/Train Loss: 0.1351/0.0010 | Test/Train Loss: 0.6447/0.0264 |
| Cross-channel | Test/Train Loss: 22.9671/0.0023 | Test/Train Loss: 23.6506/0.0375 |

## 6 CONCLUSIONS

In this paper, we showed that memorization can contribute more to generalization and out-of-distribution generalization when the model has composition capability. Our theory builds on a simple linear setting and introduces new techniques for handling long-tail features. Empirically, we show that similar intuition extends to nonlinear settings, and whether models have composition capability partly depend on their architecture. The main limitation of our work is that the theoretical result

is restricted to linear setting, while the experiments are on small networks for supervised learning problems. Our focus is on the positive role of memorization in enabling composition and illustrating when this occurs. There are also cases where memorization can hurt composition, for example when long-tail features are spuriously correlated with other features or labels, or when they are mislabeled. We hope this work serves as a starting point for understanding memorization and composition in more complicated models that go beyond supervised learning.

## ACKNOWLEDGEMENT

This work is supported by NSF Award DMS-2031849. MZ also acknowledges the support of NSF TRIPODS II DMS 2023166.

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

## A  OMITTED PROOF FOR NOISELESS RESULTS

In this section we give the detailed proof for the noiseless results Theorem 2 and Theorem 3.

**Theorem 2** (Test loss). *Suppose data generated from* (1) *with noise* $\sigma = 0$, *then with probability at least 0.99 over the data generation process, the following hold with* $k$ *satisfying* $np_k = \Theta(\ln^2 d)$ *and* $p_{>k} \leq 1 - c_p$ *with any constant* $c_p$:

$$\mathbb{E}_{x \sim \mathcal{D}}[(\hat{\boldsymbol{\beta}}^\top \boldsymbol{x} - \boldsymbol{\beta}^{*\top} \boldsymbol{x})^2] \lesssim p_{>k} + \frac{k \ln^4 d}{n^2 p_{>k}} + \frac{\ln^4 d}{n} + \frac{k p_{>k}^2 \ln^4 d}{n} + p_{>k}^3 \ln^4 d.$$

*When data follows power law decay* (2) *with* $\alpha = 1 + c_\alpha$ *with any constant* $c_\alpha > 0$

$$\mathbb{E}_{x \sim \mathcal{D}}[(\hat{\boldsymbol{\beta}}^\top \boldsymbol{x} - \boldsymbol{\beta}^{*\top} \boldsymbol{x})^2] \lesssim s \left( \frac{\ln^2 d}{ns} \right)^{1 - \frac{1}{\alpha}}$$

**Theorem 3** (OOD performance). *Under same condition as Theorem 2, then with probability at least 0.99 over the data generation process, there is a feature set* $\hat{\mathcal{F}} \subseteq \mathcal{F}$ *with size* $|\hat{\mathcal{F}}| \geq (1 - \Theta(\max\{p_k/p_{>k}, p_{>k}^2 \ln^2 d\}))|\mathcal{F}|$ *that contains almost all features show up in data such that for any feature set* $\widetilde{\mathcal{F}} \subseteq \hat{\mathcal{F}}$ *we have*

$$\mathbb{E}_{x \sim \mathcal{D}_{\widetilde{\mathcal{F}}}}[(\hat{\boldsymbol{\beta}}^\top \boldsymbol{x} - \boldsymbol{\beta}^{*\top} \boldsymbol{x})^2] \lesssim p_{>k} + \frac{k \ln^4 d}{n^2 p_{>k}} + \frac{\ln^4 d}{n} + \frac{k p_{>k}^2 \ln^4 d}{n} + p_{>k}^3 \ln^4 d.$$

*When data follows power law decay* (2) *with* $\alpha = 1 + c_\alpha$ *with any constant* $c_\alpha > 0$ *we have* $|\hat{\mathcal{F}}| \geq (1 - \Theta \left( \max \left\{ \left( \frac{\ln^2 d}{ns} \right)^{\frac{1}{\alpha}}, \frac{s^{\frac{2}{\alpha}} (\ln d)^{6 - \frac{4}{\alpha}}}{n^{2 - \frac{2}{\alpha}}} \right\} \right))|\mathcal{F}|$ *and*

$$\mathbb{E}_{x \sim \mathcal{D}_{\widetilde{\mathcal{F}}}}[(\hat{\boldsymbol{\beta}}^\top \boldsymbol{x} - \boldsymbol{\beta}^{*\top} \boldsymbol{x})^2] \lesssim s \left( \frac{\ln^2 d}{ns} \right)^{1 - \frac{1}{\alpha}}.$$

### A.1  PROOF OVERVIEW

We first show the structure of data $\boldsymbol{X}$ as below: most of data has only common features; for the data that contains long-tail features, most of them has only one long-tail feature. The proof is based on column/row-wise concentration to exploit the combinatorial structure of $\boldsymbol{X}$ instead of the concentration of whole matrix, which is difficult to handle due to high variance.

**Lemma A.1** (Structure of $\boldsymbol{X}$). *Suppose data generated from* (1), *with probability at least 0.99, the following hold with* $k$ *satisfying* $np_k = \Theta(\ln^2 d)$ *and* $p_{>k} \leq 1 - c_p$ *with any constant* $c_p$:

1. *There exist at least* $\Theta(n(1 - p_{>k}))$ *samples whose non-zero entries are only within first* $k$ *entries. That is*

$$|S_0| := |\{i \in [n] : (\boldsymbol{x}_i)_j = 0, \forall j > k\}| \geq n(1 - p_{>k}) - O(\sqrt{n(1 - p_{>k})}).$$

   *Moreover, the dimension spans by these samples in* $S_0$ *is* $k$.

2. *There exist at least* $\Theta(np_{>k}(1 - p_{>k}))$ *samples that only have exactly one non-zero entry in long-tail entries (not in the first* $k$ *entries). That is*

$$|S_1| := |\{i \in [n] : \exists \ell > k \text{ such that } (\boldsymbol{x}_i)_\ell \neq 0 \text{ and } (\boldsymbol{x}_i)_j = 0, \forall j > k, j \neq \ell\}|$$
$$\geq np_{>k}(1 - p_{>k}) - O(\sqrt{np_{>k}(1 - p_{>k})})$$

   *and* $|S_1| \lesssim np_{>k}$.

3. *Denote the set of features show up in data as*

$$\mathcal{F} = \{j \in [d] : \exists i \in [n] s.t. (\boldsymbol{x}_i)_j \neq 0\}.$$

   *Similarly define* $\mathcal{F}_0$ *and* $\mathcal{F}_1$ *for* $S_0$ *and* $S_1$ *as the set of features show up in* $S_0$ *and* $S_1$.

   *We have total number of features shown up in data is* $|\mathcal{F}| = k + \Theta(np_{>k})$ *and number of long tail features is* $|\mathcal{F}_1 \setminus \mathcal{F}_0| \geq \Theta(p_{>k}/(p_k \ln^2 d))$. *Moreover, the number of features show up in* $S_0$ *and* $S_1$ *is at least*

$$|\mathcal{F}_0 \cup \mathcal{F}_1| \geq \left( 1 - \Theta \left( \max\{p_k/p_{>k}, p_{>k}^2 \ln^2 d\} \right) \right) |\mathcal{F}|$$

Given the structure of the data $X$, we obtain the following parameter recovery guarantee. Intuitively, since most of the data contains only the common feature, the common part is uniquely determined. Then, for the data points that contain only one long-tail feature, the long-tail part can also be uniquely recovered once common part is recovered. Together, these observations lead to the parameter recovery result presented below.

**Lemma A.2** (Recovery of $\beta^*$). *Suppose data generated from* (1) *with noise* $\sigma = 0$, *for k that satisfies the condition of Lemma A.1, we have*

1. *Common features (Top-k entries) are recovered:* $\hat{\boldsymbol{\beta}}_{\leq k} = \boldsymbol{\beta}^*_{\leq k}$

2. *Most of long tail features shown up in data X(last-$d - k$ entries) are recovered:*

$$\frac{\left|\left\{i \in [d] : \hat{\beta}_i = \beta_i^*\right\}\right|}{|\mathcal{F}|} \geq 1 - \Theta(\max\{p_k/p_{>k}, p_{>k}^2 \ln^2 d\}).$$

3. $\left\|\hat{\boldsymbol{\beta}}\right\|_2 \leq \|\boldsymbol{\beta}^*\|_2$ *and* $\hat{\beta}_i = 0$ *for* $i \notin \mathcal{F}$.

As a corollary, we choose the threshold $k$ under our data generation process (2) to get the following result.

**Corollary A.3.** *Suppose data generated from* (1) *with noise* $\sigma = 0$ *and follows power law decay* (2) *with* $\alpha = 1 + c_\alpha$ *for any constant* $c_\alpha > 0$ , *then we have for* $k = \Theta((ns/\ln^2 d)^{1/\alpha})$ *(i.e.* $np_k = \Theta(\ln^2 d)$)

1. *Common features (Top-k entries) are recovered:* $\hat{\boldsymbol{\beta}}_{\leq k} = \boldsymbol{\beta}^*_{\leq k}$

2. *Most of long tail features shown up in data X (last-$d - k$ entries) are recovered:*

$$\frac{\left|\left\{i \in [d] : \hat{\beta}_i = \beta_i^*\right\}\right|}{|\mathcal{F}|} \geq 1 - \Theta\left(\max\left\{\left(\frac{\ln^2 d}{ns}\right)^{\frac{1}{\alpha}}, \frac{s^{\frac{2}{\alpha}}(\ln d)^{6 - \frac{4}{\alpha}}}{n^{2 - \frac{2}{\alpha}}}\right\}\right).$$

3. $\left\|\hat{\boldsymbol{\beta}}\right\|_2 \leq \|\boldsymbol{\beta}^*\|_2$ *and* $\hat{\beta}_i = 0$ *for* $i \notin \mathcal{F}$.

*Proof.* The proof follows from Lemma A.2 and Claim 1 by choosing $k = \Theta((ns/\ln^2 d)^{1/\alpha})$ (i.e., $np_k = \Theta(\ln^2 d)$). $\qquad\square$

Now, given all parameter recovery results above, we can get the test loss and OOD loss guarantee. We give the details in the next section.

## A.2 PROOF OF MAIN RESULTS

Building on the results from the previous section, we are now ready to present the proofs of the main results. Both proofs mostly follow from the parameter recovery guarantee in Lemma A.2 and its specialization to power-law data in Corollary A.3.

**Theorem 2** (Test loss). *Suppose data generated from* (1) *with noise* $\sigma = 0$, *then with probability at least 0.99 over the data generation process, the following hold with k satisfying* $np_k = \Theta(\ln^2 d)$ *and* $p_{>k} \leq 1 - c_p$ *with any constant* $c_p$:

$$\mathbb{E}_{x \sim \mathcal{D}}[(\hat{\boldsymbol{\beta}}^\top \boldsymbol{x} - \boldsymbol{\beta}^{*\top} \boldsymbol{x})^2] \lesssim p_{>k} + \frac{k \ln^4 d}{n^2 p_{>k}} + \frac{\ln^4 d}{n} + \frac{kp_{>k}^2 \ln^4 d}{n} + p_{>k}^3 \ln^4 d.$$

*When data follows power law decay* (2) *with* $\alpha = 1 + c_\alpha$ *with any constant* $c_\alpha > 0$

$$\mathbb{E}_{x \sim \mathcal{D}}[(\hat{\boldsymbol{\beta}}^\top \boldsymbol{x} - \boldsymbol{\beta}^{*\top} \boldsymbol{x})^2] \lesssim s\left(\frac{\ln^2 d}{ns}\right)^{1 - \frac{1}{\alpha}}$$

*Proof.* From Lemma A.2 we know

$$\mathbb{E}_{x \sim \mathcal{D}}[(\hat{\boldsymbol{\beta}}^\top \boldsymbol{x} - \boldsymbol{\beta}^{*\top} \boldsymbol{x})^2] = \sum_i p_i (\hat{\beta}_i - \beta_i^*)^2 \lesssim \sum_{i>k} p_i (\beta_i^*)^2 + p_k |\mathcal{F} \setminus (\mathcal{F}_0 \cup \mathcal{F}_1)|$$

$$\lesssim p_{>k} + \frac{k \ln^4 d}{n^2 p_{>k}} + \frac{\ln^4 d}{n} + \frac{k p_{>k}^2 \ln^4 d}{n} + p_{>k}^3 \ln^4 d.$$

When data follows power law decay (2), we know $k = \Theta((ns/\ln d)^{1/\alpha})$ from $np_k = \Theta(\ln d)$. Then from Claim 1 that $p_{>k} \lesssim s \left(\frac{\ln d}{ns}\right)^{1 - \frac{1}{\alpha}}$ and get the result. $\qquad \square$

**Theorem 3** (OOD performance). *Under same condition as Theorem 2, then with probability at least 0.99 over the data generation process, there is a feature set $\hat{\mathcal{F}} \subseteq \mathcal{F}$ with size $|\hat{\mathcal{F}}| \geq (1 - \Theta(\max\{p_k/p_{>k}, p_{>k}^2 \ln^2 d\}))|\mathcal{F}|$ that contains almost all features show up in data such that for any feature set $\widetilde{\mathcal{F}} \subseteq \hat{\mathcal{F}}$ we have*

$$\mathbb{E}_{x \sim \mathcal{D}_{\widetilde{\mathcal{F}}}}[(\hat{\boldsymbol{\beta}}^\top \boldsymbol{x} - \boldsymbol{\beta}^{*\top} \boldsymbol{x})^2] \lesssim p_{>k} + \frac{k \ln^4 d}{n^2 p_{>k}} + \frac{\ln^4 d}{n} + \frac{k p_{>k}^2 \ln^4 d}{n} + p_{>k}^3 \ln^4 d.$$

*When data follows power law decay (2) with $\alpha = 1 + c_\alpha$ with any constant $c_\alpha > 0$ we have $|\hat{\mathcal{F}}| \geq (1 - \Theta\left(\max\left\{\left(\frac{\ln^2 d}{ns}\right)^{\frac{1}{\alpha}}, \frac{s^{\frac{2}{\alpha}}(\ln d)^{6 - \frac{4}{\alpha}}}{n^{2 - \frac{2}{\alpha}}}\right\}\right))|\mathcal{F}|$ and*

$$\mathbb{E}_{x \sim \mathcal{D}_{\widetilde{\mathcal{F}}}}[(\hat{\boldsymbol{\beta}}^\top \boldsymbol{x} - \boldsymbol{\beta}^{*\top} \boldsymbol{x})^2] \lesssim s \left(\frac{\ln^2 d}{ns}\right)^{1 - \frac{1}{\alpha}}.$$

*Proof.* It suffices to let $\hat{\mathcal{F}} = \mathcal{F}_0 \cup \mathcal{F}_1$. From Lemma A.2 we know the bound on size $|\hat{\mathcal{F}}|$ and

$$\mathbb{E}_{x \sim \mathcal{D}_{\widetilde{\mathcal{F}}}}[(\hat{\boldsymbol{\beta}}^\top \boldsymbol{x} - \boldsymbol{\beta}^{*\top} \boldsymbol{x})^2] = \sum_{i \notin \widetilde{\mathcal{F}}} p_i (\hat{\beta}_i - \beta_i^*)^2 + \sum_{i \in \widetilde{\mathcal{F}}} (\hat{\beta}_i - \beta_i^*)^2 \leq \sum_{i > k, i \notin \mathcal{F}_0 \cup \mathcal{F}_1} p_i (\beta_i^*)^2$$

$$\lesssim p_{>k} + \frac{k \ln^4 d}{n^2 p_{>k}} + \frac{\ln^4 d}{n} + \frac{k p_{>k}^2 \ln^4 d}{n} + p_{>k}^3 \ln^4 d.$$

where the calculation is same as in Theorem 2.

When data follows power law decay (2), similar as in the proof of Theorem 2 we know the bound. Size of $|\hat{\mathcal{F}}|$ follows from Lemma A.3. $\qquad \square$

## A.3 DETAILED PROOFS

Here we give the omitted proofs in Appendix A.1.

**Lemma A.1** (Structure of $\boldsymbol{X}$). *Suppose data generated from (1), with probability at least 0.99, the following hold with $k$ satisfying $np_k = \Theta(\ln^2 d)$ and $p_{>k} \leq 1 - c_p$ with any constant $c_p$:*

1. *There exist at least $\Theta(n(1 - p_{>k}))$ samples whose non-zero entries are only within first $k$ entries. That is*

$$|S_0| := |\{i \in [n] : (\boldsymbol{x}_i)_j = 0, \forall j > k\}| \geq n(1 - p_{>k}) - O(\sqrt{n(1 - p_{>k})}).$$

   *Moreover, the dimension spans by these samples in $S_0$ is $k$.*

2. *There exist at least $\Theta(np_{>k}(1 - p_{>k}))$ samples that only have exactly one non-zero entry in long-tail entries (not in the first $k$ entries). That is*

$$|S_1| := |\{i \in [n] : \exists \ell > k \text{ such that } (\boldsymbol{x}_i)_\ell \neq 0 \text{ and } (\boldsymbol{x}_i)_j = 0, \forall j > k, j \neq \ell\}|$$
$$\geq np_{>k}(1 - p_{>k}) - O(\sqrt{np_{>k}(1 - p_{>k})})$$

   *and $|S_1| \lesssim np_{>k}$.*

3. *Denote the set of features show up in data as*

$$\mathcal{F} = \{j \in [d] : \exists i \in [n] s.t. (\boldsymbol{x}_i)_j \neq 0\}.$$

*Similarly define $\mathcal{F}_0$ and $\mathcal{F}_1$ for $S_0$ and $S_1$ as the set of features show up in $S_0$ and $S_1$.*

*We have total number of features shown up in data is $|\mathcal{F}| = k + \Theta(np_{>k})$ and number of long tail features is $|\mathcal{F}_1 \setminus \mathcal{F}_0| \geq \Theta(p_{>k}/(p_k \ln^2 d))$. Moreover, the number of features show up in $S_0$ and $S_1$ is at least*

$$|\mathcal{F}_0 \cup \mathcal{F}_1| \geq \left(1 - \Theta\left(\max\{p_k/p_{>k}, p_{>k}^2 \ln^2 d\}\right)\right)|\mathcal{F}|$$

*Proof.* We show one by one.

**item 1**  For any data $\boldsymbol{x} \in \mathbb{R}^d$ generated by (2), we first have the probability

$$\mathbb{P}\left(x_i = 0, \forall i > k\right) = \prod_{i>k}(1 - p_i) \geq 1 - p_{>k},$$

where we use Lemma C.1.

Since each data is generated i.i.d. following (2), by standard Chernoff's bound, we know with probability at least 0.999

$$|S_0| = |\{i \in [n] : (\boldsymbol{x}_i)_j = 0, \forall j > k\}| \geq n(1 - p_{>k}) - O(\sqrt{n(1 - p_{>k})}).$$

As for the dimension, since $np_k \gtrsim \ln d$, we can show $\boldsymbol{X}_{S_0,\leq k}^\top \boldsymbol{X}_{S_0,\leq k}$ has full rank as in Lemma C.2, where $\boldsymbol{X}_{S_0,\leq k}$ is the data matrix $\boldsymbol{X} \in \mathbb{R}^{n \times d}$ constrained with only rows in $S_0$ and first $k$ columns.

**item 2**  Similar as item 1, for any data $\boldsymbol{x} \in \mathbb{R}^d$ generated by (2), we first have the probability

$$\mathbb{P}\left(\exists \ell > k \text{ such that } x_\ell \neq 0 \text{ and } x_j = 0, \forall j > k, j \neq \ell\right)$$
$$= \sum_{\ell > k} p_\ell \prod_{j>k,j\neq\ell}(1 - p_i) \geq \sum_{\ell > k} p_\ell(1 - p_{>k} + p_\ell) \geq p_{>k}(1 - p_{>k}),$$

where we use Lemma C.1. We can also see the above probability is at most $p_{>k}$.

Since each data is generated i.i.d. following (2), by standard Chernoff's bound, we know with probability at least 0.999

$$|S_1| = |\{i \in [n] : \exists \ell > k \text{ such that } (\boldsymbol{x}_i)_\ell \neq 0 \text{ and } (\boldsymbol{x}_i)_j = 0, \forall j > k, j \neq \ell\}|$$
$$\geq np_{>k}(1 - p_{>k}) - O(\sqrt{np_{>k}(1 - p_{>k})})$$

and $|S_1| \lesssim np_{>k}$.

**item 3**  The total number of features show up in data is upper bounded by $k + \|\boldsymbol{X}_{>k}\|_F^2$ (top-$k$ entries and all non-zero entries in the remaining entries). Thus, we know $|\mathcal{F}| = k + \Theta(np_{>k})$ by Chernoff's bound.

For every feature not in top-$k$, it at most shows up in $\Theta(np_i + \sqrt{np_i}) = O(\ln^2 d)$ data by Chernoff's bound and $np_i \leq np_k = \Theta(\ln^2 d)$. Therefore, the total number of features show up in $S_0$ and $S_1$ is at least $|\mathcal{F}_0 \cup \mathcal{F}_1| \geq k + |S_1|/O(\ln^2 d) \geq k + \Theta(np_{>k}/\ln^2 d)$. So number of long tail features is at least $|\mathcal{F}_1 \setminus \mathcal{F}_0| \geq \Theta(np_{>k}/\ln^2 d)$.

On the other hand, from item 1 and 2 we can see the probability of data not in $S_0 \cup S_1$ is at most $p_{>k}^2$, so the total number of such data is at most $\Theta(\max\{1, np_{>k}^2\})$ by Chernoff's bound. Thus, the total nonzero entries (not include top-$k$ entries) in these data is at most $\Theta(\max\{1, \max\{1, np_{>k}^2\} \cdot p_{>k}\}) = \Theta(\max\{1, np_{>k}^3\})$. This implies the number of features show up not in $S_0$ and $S_1$ is at most $|\mathcal{F} \setminus \mathcal{F}_0 \cup \mathcal{F}_1| \leq \Theta(\max\{1, np_{>k}^3\})$. Hence, the number of features show up in $S_0$ and $S_1$ is at least

$$|\mathcal{F}_0 \cup \mathcal{F}_1| = |\mathcal{F}|\frac{1}{1 + |\mathcal{F} \setminus \mathcal{F}_0 \cup \mathcal{F}_1|/|\mathcal{F}_0 \cup \mathcal{F}_1|} \geq |\mathcal{F}|\frac{1}{1 + \Theta(\max\{p_k/p_{>k}, p_{>k}^2 \ln^2 d\})}$$
$$= |\mathcal{F}|(1 - \Theta(\max\{p_k/p_{>k}, p_{>k}^2 \ln^2 d\})).$$

$\square$

**Lemma A.2** (Recovery of $\beta^*$). *Suppose data generated from* (1) *with noise* $\sigma = 0$, *for* $k$ *that satisfies the condition of Lemma A.1, we have*

1. *Common features (Top-$k$ entries) are recovered:* $\hat{\boldsymbol{\beta}}_{\leq k} = \boldsymbol{\beta}^*_{\leq k}$

2. *Most of long tail features shown up in data $X$(last-$d - k$ entries) are recovered:*

$$\frac{\left|\left\{i \in [d] : \hat{\beta}_i = \beta_i^*\right\}\right|}{|\mathcal{F}|} \geq 1 - \Theta(\max\{p_k/p_{>k}, p_{>k}^2 \ln^2 d\}).$$

3. $\left\|\hat{\boldsymbol{\beta}}\right\|_2 \leq \|\boldsymbol{\beta}^*\|_2$ *and* $\hat{\beta}_i = 0$ *for* $i \notin \mathcal{F}$.

*Proof.* We prove one-by-one. The results are mostly a combination of Lemma A.1 and Claim 1.

**item 1** For common features (top-$k$ entries), it suffices to only look at data in $S_0$, that is data only have non-zero entries in top-$k$ entries. From Lemma A.1, we know the only solution is $\hat{\boldsymbol{\beta}}_{\leq k} = \boldsymbol{\beta}^*_{\leq k}$.

**item 2** For long tail features, we only focus on those show up in $S_1$ (data only have 1 non-zero entries in tail entries). From item 1, we know top-$k$ entries are recovered. Thus, the only solution to data in $S_1$ is $\hat{\beta}_i = \beta_i^*$ for $i \in \mathcal{F}_1$. From Lemma A.1, we know $\mathcal{F}_1 \cup \mathcal{F}_1$ covers most of the features show up in data: $|\mathcal{F}_0 \cup \mathcal{F}_1| \geq (1 - \Theta(\max\{p_k/p_{>k}, p_{>k}^2 \ln^2 d\})|\mathcal{F}|$. Thus, from above we know the $\beta_i^*$ for $i \in \mathcal{F}_0 \cup \mathcal{F}_1$ are recovered.

**item 3** This is directly implied by the meaning of min-$\ell_2$-norm interpolator. $\square$

## B  PROOFS FOR NOISY CASE

In this section, we give the proofs for the noisy case results as below:

**Theorem 4** (Test loss). *Suppose data generated from* (1) *with noise* $\sigma \leq 1$, *with probability at least 0.99 over the data generation process, the following hold with $k$ satisfying $np_k = \Theta(\ln^2 d)$, $p_{>k} \leq 1 - c_p$ with any constant $c_p$ and $np_{>k}^2 < c$ for a small enough constant $c$:*

$$\mathbb{E}_{\boldsymbol{X}_{\leq k}}[\mathbb{E}_{x \sim \mathcal{D}}[(\hat{\boldsymbol{\beta}}^\top \boldsymbol{x} - \boldsymbol{\beta}^{*\top} \boldsymbol{x})^2]] \lesssim p_{>k} + \sigma^2 \left(\frac{k \ln d}{n} + \left(\frac{k^2 \ln^2 d}{n} + \ln d\right) p_{>k} \ln d\right)$$

*where the first expectation is over data generating process of common part of training data $\boldsymbol{X}_{\leq k}$.*

*When data follows power law decay* (2) *with* $\alpha = 2 + c_\alpha$ *with any constant* $c_\alpha > 0$

$$\mathbb{E}_{\boldsymbol{X}_{\leq k}}[\mathbb{E}_{x \sim \mathcal{D}}[(\hat{\boldsymbol{\beta}}^\top \boldsymbol{x} - \boldsymbol{\beta}^{*\top} \boldsymbol{x})^2]] \lesssim s \left(\frac{\ln^2 d}{ns}\right)^{1 - \frac{1}{\alpha}} + \sigma^2 \left(\frac{\ln^2 d}{ns}\right)^{1 - \frac{1}{\alpha}} s \ln^5 d$$

**Theorem 5** (OOD performance). *Under same condition as Theorem 4, then with probability at least 0.99 over the data generation process, consider any feature set $\widetilde{\mathcal{F}} \subseteq \mathcal{F} \setminus \mathcal{F}_0$ contains any long tail features show up in data, we have*

$$\mathbb{E}_{\boldsymbol{X}_{\leq k}}[\mathbb{E}_{x \sim \mathcal{D}_{\widetilde{\mathcal{F}}}}[(\hat{\boldsymbol{\beta}}^\top \boldsymbol{x} - \boldsymbol{\beta}^{*\top} \boldsymbol{x})^2]] \lesssim p_{>k} + \sigma^2 |\widetilde{\mathcal{F}}| \left(\frac{k^2 \ln^2 d}{n} + \ln d\right),$$

*where the first expectation is over data generating process of common part of training data $\boldsymbol{X}_{\leq k}$.*

*When data follows power law decay* (2) *with* $\alpha = 2 + c_\alpha$ *with any constant* $c_\alpha > 0$

$$\mathbb{E}_{\boldsymbol{X}_{\leq k}}[\mathbb{E}_{x \sim \mathcal{D}}[(\hat{\boldsymbol{\beta}}^\top \boldsymbol{x} - \boldsymbol{\beta}^{*\top} \boldsymbol{x})^2]] \lesssim s \left(\frac{\ln^2 d}{ns}\right)^{1 - \frac{1}{\alpha}} + \sigma^2 |\widetilde{\mathcal{F}}| \ln d.$$

## B.1 PROOF OVERVIEW

Here we give the proof overview of noisy case results.

Besides the structure of data $X$ as described in Lemma A.1, we have the additional following structure under a stronger tail decay assumption. Now we can show all data has either 0 or 1 long tail feature.

**Lemma B.1.** *Under Lemma A.1 with additional requirement $np_{>k}^2 < c$ for a small enough constant $c$, then besides the conclusion of Lemma A.1 we additionally have there is no combination of long tail features:*

$$|S_{\geq 2}| = |\{i \in [n] : \exists \ell_1 \neq \ell_2 > k \text{ such that } x_{\ell_1}, x_{\ell_2} \neq 0\}| = 0.$$

Given the structure of the data $X$, we can establish the following parameter recovery guarantee. Intuitively, the common features can be recovered accurately, since most of the data contains only common features (similar to the noiseless case). For the long-tail features, consider the simplest scenario where a feature appears only once in the entire dataset. In that case, the corresponding sample must have zero training loss, so the feature can be recovered up to the noise level $\sigma$. The general case follows a similar argument: we can control the training loss for samples that contain a given long-tail feature.

**Lemma B.2** (Recovery of $\beta^*$)**.** *Under Lemma A.1 and Lemma B.1, with probability 0.99 we have (in below expectation is over data generating process of common part of training data $X_{\leq k}$)*

1. *Common features (Top-$k$ entries) are recovered:*

$$\mathbb{E}_{X_{\leq k}} \left[ \left\| \hat{\beta}_{\leq k} - \beta^*_{\leq k} \right\|^2_{\Sigma_{\leq k}} \right] \lesssim \sigma^2 \left( \frac{k}{n} + p_{>k} \right).$$

2. *All long tail features shown up in data X(last-$d-k$ entries) are recovered upto noise level:*

$$\mathbb{E}_{X_{\leq k}}[|\hat{\beta}_i - \beta^*_i|^2] \lesssim \sigma^2 \left( \frac{k^2 \ln^2 d}{n} + kp_{>k} + \ln d \right).$$

3. $\hat{\beta}_i = 0$ *for all $i \notin (\mathcal{F}_0 \cup \mathcal{F}_1)$.*

As a corollary, we choose the threshold $k$ under our data generation process (2) to get the following result.

**Corollary B.3.** *For data generating process (2) of power law decay, with $\alpha = 2 + c_\alpha$ with any constant $c_\alpha > 0$, we have for $k = \Theta((ns/\ln^2 d)^{1/\alpha})$ (i.e. $np_k = \Theta(\ln^2 d)$)*

1. *Common features (Top-$k$ entries) are recovered:*

$$\mathbb{E}_{X_{\leq k}} \left[ \left\| \hat{\beta}_{\leq k} - \beta^*_{\leq k} \right\|^2_{\Sigma_{\leq k}} \right] \lesssim \sigma^2 \frac{s^{\frac{1}{\alpha}} (\ln d)^{2 - \frac{1}{\alpha}}}{n^{1 - \frac{1}{\alpha}}}.$$

2. *All long tail features shown up in data X(last-$d-k$ entries) are recovered upto noise level:*

$$\mathbb{E}_{X_{\leq k}}[|\hat{\beta}_i - \beta^*_i|^2] \lesssim \sigma^2 \ln d,$$

*where expectation is over data generating process of training data.*

3. $\hat{\beta}_i = 0$ *for all $i \notin (\mathcal{F}_0 \cup \mathcal{F}_1)$.*

*Proof.* The proof follows from Lemma B.2 and Claim 1 by choosing $k = \Theta((ns/\ln^2 d)^{1/\alpha})$ (i.e., $np_k = \Theta(\ln^2 d)$). $\square$

## B.2 PROOF OF MAIN RESULTS

Building on the results from the previous section, we are now ready to present the proofs of the main results. Both proofs mostly follow from the parameter recovery guarantee in Lemma B.2 and its specialization to power-law data in Corollary B.3.

**Theorem 4** (Test loss). *Suppose data generated from (1) with noise $\sigma \leq 1$, with probability at least 0.99 over the data generation process, the following hold with $k$ satisfying $np_k = \Theta(\ln^2 d)$, $p_{>k} \leq 1 - c_p$ with any constant $c_p$ and $np_{>k}^2 < c$ for a small enough constant $c$:*

$$\mathbb{E}_{\boldsymbol{X}_{\leq k}}[\mathbb{E}_{x \sim \mathcal{D}}[(\hat{\boldsymbol{\beta}}^\top \boldsymbol{x} - \boldsymbol{\beta}^{*\top} \boldsymbol{x})^2]] \lesssim p_{>k} + \sigma^2 \left( \frac{k \ln d}{n} + \left( \frac{k^2 \ln^2 d}{n} + \ln d \right) p_{>k} \ln d \right)$$

*where the first expectation is over data generating process of common part of training data $\boldsymbol{X}_{\leq k}$.*

*When data follows power law decay (2) with $\alpha = 2 + c_\alpha$ with any constant $c_\alpha > 0$*

$$\mathbb{E}_{\boldsymbol{X}_{\leq k}}[\mathbb{E}_{x \sim \mathcal{D}}[(\hat{\boldsymbol{\beta}}^\top \boldsymbol{x} - \boldsymbol{\beta}^{*\top} \boldsymbol{x})^2]] \lesssim s \left( \frac{\ln^2 d}{ns} \right)^{1 - \frac{1}{\alpha}} + \sigma^2 \left( \frac{\ln^2 d}{ns} \right)^{1 - \frac{1}{\alpha}} s \ln^5 d$$

*Proof.* From Lemma B.2 we know

$$\mathbb{E}_{\boldsymbol{X}_{\leq k}}[\mathbb{E}_{x \sim \mathcal{D}}[(\hat{\boldsymbol{\beta}}^\top \boldsymbol{x} - \boldsymbol{\beta}^{*\top} \boldsymbol{x})^2]] = \sum_i p_i \mathbb{E}_{\boldsymbol{X}_{\leq k}}[(\hat{\beta}_i - \beta_i^*)^2]$$

$$\lesssim \sigma^2 \left( \frac{k}{n} + p_{>k} \right) + \sum_{i>k} p_i (\beta_i^*)^2 + \sigma^2 \left( \frac{k^2 \ln^2 d}{n} + kp_{>k} + \ln d \right) p_k |\mathcal{F}_1 \setminus \mathcal{F}_0|$$

$$\lesssim p_{>k} + \sigma^2 \left( \frac{k \ln d}{n} + \left( \frac{k^2 \ln^2 d}{n} + \ln d \right) p_{>k} \ln^2 d \right),$$

where we use the condition on $p_{>k}$ to simplify the expression.

When data follows power law decay (2), we know $k = \Theta((ns/\ln^2 d)^{1/\alpha})$ from $np_k = \Theta(\ln^2 d)$. Then from Claim 1 that $p_{>k} \lesssim s \left( \frac{\ln^2 d}{ns} \right)^{1 - \frac{1}{\alpha}}$ and $\frac{k \ln d}{n} + \left( \frac{k^2 \ln^2 d}{n} + \ln d \right) p_{>k} \ln^2 d \lesssim \frac{s^{\frac{1}{\alpha}} (\ln d)^{7 - \frac{2}{\alpha}}}{n^{1 - \frac{1}{\alpha}}}$. $\square$

**Theorem 5** (OOD performance). *Under same condition as Theorem 4, then with probability at least 0.99 over the data generation process, consider any feature set $\widetilde{\mathcal{F}} \subseteq \mathcal{F} \setminus \mathcal{F}_0$ contains any long tail features show up in data, we have*

$$\mathbb{E}_{\boldsymbol{X}_{\leq k}}[\mathbb{E}_{x \sim \mathcal{D}_{\widetilde{\mathcal{F}}}}[(\hat{\boldsymbol{\beta}}^\top \boldsymbol{x} - \boldsymbol{\beta}^{*\top} \boldsymbol{x})^2]] \lesssim p_{>k} + \sigma^2 |\widetilde{\mathcal{F}}| \left( \frac{k^2 \ln^2 d}{n} + \ln d \right),$$

*where the first expectation is over data generating process of common part of training data $\boldsymbol{X}_{\leq k}$.*

*When data follows power law decay (2) with $\alpha = 2 + c_\alpha$ with any constant $c_\alpha > 0$*

$$\mathbb{E}_{\boldsymbol{X}_{\leq k}}[\mathbb{E}_{x \sim \mathcal{D}}[(\hat{\boldsymbol{\beta}}^\top \boldsymbol{x} - \boldsymbol{\beta}^{*\top} \boldsymbol{x})^2]] \lesssim s \left( \frac{\ln^2 d}{ns} \right)^{1 - \frac{1}{\alpha}} + \sigma^2 |\widetilde{\mathcal{F}}| \ln d.$$

*Proof.* From Lemma B.2 we know

$$\mathbb{E}[\mathbb{E}_{x \sim \mathcal{D}_{\widetilde{\mathcal{F}}}}[(\hat{\boldsymbol{\beta}}^\top \boldsymbol{x} - \boldsymbol{\beta}^{*\top} \boldsymbol{x})^2]] = \sum_{i \notin \widetilde{\mathcal{F}}} p_i \mathbb{E}[(\hat{\beta}_i - \beta_i^*)^2] + \sum_{i \in \widetilde{\mathcal{F}}} \mathbb{E}[(\hat{\beta}_i - \beta_i^*)^2]$$

$$\lesssim \sigma^2 \left( \frac{k}{n} + p_{>k} \right) + \sum_{i>k} p_i (\beta_i^*)^2 + \left( p_k |\mathcal{F}_1 \setminus (\widetilde{\mathcal{F}} \cup \mathcal{F}_0)| + |\widetilde{\mathcal{F}}| \right) \sigma^2 \left( \frac{k^2 \ln^2 d}{n} + kp_{>k} + \ln d \right)$$

$$\lesssim p_{>k} + \sigma^2 |\widetilde{\mathcal{F}}| \left( \frac{k^2 \ln^2 d}{n} + \ln d \right).$$

When data follows power law decay (2), we know $k = \Theta((ns/\ln^2 d)^{1/\alpha})$ from $np_k = \Theta(\ln^2 d)$. Then from Claim 1 we get the result. $\square$

## B.3 DETAILED PROOFS

Here we present the detailed proofs in Appendix B.1

**Lemma B.1.** *Under Lemma A.1 with additional requirement $np_{>k}^2 < c$ for a small enough constant c, then besides the conclusion of Lemma A.1 we additionally have there is no combination of long tail features:*

$$|S_{\geq 2}| = |\{i \in [n] : \exists \ell_1 \neq \ell_2 > k \text{ such that } x_{\ell_1}, x_{\ell_2} \neq 0\}| = 0.$$

*Proof.* From the proof of Lemma A.1 we know the probability of one data has at least 2 long tail features is

$$\mathbb{P} \left( \exists \ell_1 \neq \ell_2 > k \text{ such that } x_{\ell_1}, x_{\ell_2} \neq 0 \right)$$
$$= 1 - \mathbb{P} \left( x_i = 0, \forall i > k \right) - \mathbb{P} \left( \exists \ell > k \text{ such that } x_\ell \neq 0 \text{ and } x_j = 0, \forall j > k, j \neq \ell \right)$$
$$\leq 1 - p_{>k} - p_{>k}(1 - p_{>k}) = p_{>k}^2,$$

Since each data is generated i.i.d. following (2), by standard Chernoff's bound, we know with probability at least 0.999

$$|S_{\geq 2}| = |\{i \in [n] : \exists \ell_1 \neq \ell_2 > k \text{ such that } x_{\ell_1}, x_{\ell_2} \neq 0\}| \lesssim np_{>k}^2 + \sqrt{np_{>k}^2} < 1,$$

which implies $|S_{\geq 2}| = 0$. $\qquad\qquad\square$

**Lemma B.2** (Recovery of $\beta^*$). *Under Lemma A.1 and Lemma B.1, with probability 0.99 we have (in below expectation is over data generating process of common part of training data $X_{\leq k}$)*

1. *Common features (Top-k entries) are recovered:*

$$\mathbb{E}_{\boldsymbol{X}_{\leq k}} \left[ \left\| \hat{\boldsymbol{\beta}}_{\leq k} - \boldsymbol{\beta}_{\leq k}^* \right\|_{\boldsymbol{\Sigma}_{\leq k}}^2 \right] \lesssim \sigma^2 \left( \frac{k}{n} + p_{>k} \right).$$

2. *All long tail features shown up in data X(last-d − k entries) are recovered upto noise level:*

$$\mathbb{E}_{\boldsymbol{X}_{\leq k}}[|\hat{\beta}_i - \beta_i^*|^2] \lesssim \sigma^2 \left( \frac{k^2 \ln^2 d}{n} + kp_{>k} + \ln d \right).$$

3. $\hat{\beta}_i = 0$ *for all $i \notin (\mathcal{F}_0 \cup \mathcal{F}_1)$.*

*Proof.* We prove one-by-one.

**item 1** First note that since $\hat{\boldsymbol{\beta}}$ is the minima of training loss, we know training loss is bounded $\left\| \boldsymbol{X}\hat{\boldsymbol{\beta}} - \boldsymbol{y} \right\|_2^2 \leq \|\boldsymbol{X}\boldsymbol{\beta}^* - \boldsymbol{y}\|_2^2 = \|\boldsymbol{\xi}\|_2^2$.

For common features (top-$k$ entries), it suffices to only look at data in $S_0$, that is data only have non-zero entries in top-$k$ entries. We know from above that

$$\left\| \boldsymbol{X}_{S_0, \leq k}\hat{\boldsymbol{\beta}}_{\leq k} - \boldsymbol{y}_{S_0} \right\|_2^2 \leq \|\boldsymbol{\xi}\|_2^2$$

Denote the normalized matrix $\widetilde{\boldsymbol{X}}_{S_0, \leq k} = \boldsymbol{X}_{S_0, \leq k}\boldsymbol{\Sigma}_{\leq k}^{-1/2}$. To simplify the notation within in the following equation, we omit the subscript $S_0$ and $\leq k$. We have

$$\left\| \widetilde{\boldsymbol{X}}\boldsymbol{\Sigma}^{1/2}\hat{\boldsymbol{\beta}} - \boldsymbol{y} \right\|_2^2 = (\hat{\boldsymbol{\beta}} - \boldsymbol{\beta}^*)\boldsymbol{\Sigma}^{1/2}\widetilde{\boldsymbol{X}}^\top\widetilde{\boldsymbol{X}}\boldsymbol{\Sigma}^{1/2}(\hat{\boldsymbol{\beta}} - \boldsymbol{\beta}^*) - 2\boldsymbol{\xi}^\top\widetilde{\boldsymbol{X}}\boldsymbol{\Sigma}^{1/2}(\hat{\boldsymbol{\beta}} - \boldsymbol{\beta}^*) + \|\boldsymbol{\xi}\|_2^2$$

$$\geq |S_0| \left( 1 - \frac{1}{|S_0|} \left\| \widetilde{\boldsymbol{X}}^\top\widetilde{\boldsymbol{X}} - |S_0|\boldsymbol{I} \right\|_2 \right) \left\| \hat{\boldsymbol{\beta}} - \boldsymbol{\beta}^* \right\|_{\boldsymbol{\Sigma}}^2 - 2 \left\| \boldsymbol{\xi}^\top\widetilde{\boldsymbol{X}} \right\|_2 \left\| \hat{\boldsymbol{\beta}} - \boldsymbol{\beta}^* \right\|_{\boldsymbol{\Sigma}} + \|\boldsymbol{\xi}\|_2^2$$

Rearranging the terms we have

$$|S_0| \left( 1 - \frac{1}{|S_0|} \left\| \widetilde{\boldsymbol{X}}_{S_0, \leq k}^\top\widetilde{\boldsymbol{X}}_{S_0, \leq k} - |S_0|\boldsymbol{I} \right\|_2 \right) \left\| \hat{\boldsymbol{\beta}}_{\leq k} - \boldsymbol{\beta}_{\leq k}^* \right\|_{\boldsymbol{\Sigma}_{\leq k}}^2 - 2 \left\| \boldsymbol{\xi}_{S_0}^\top\widetilde{\boldsymbol{X}}_{S_0, \leq k} \right\|_2 \left\| \hat{\boldsymbol{\beta}}_{\leq k} - \boldsymbol{\beta}_{\leq k}^* \right\|_{\boldsymbol{\Sigma}_{\leq k}}$$
$$\leq \|\boldsymbol{\xi}\|_2^2 - \|\boldsymbol{\xi}_{S_0}\|_2^2,$$

which leads to

$$\left\|\hat{\boldsymbol{\beta}}_{\leq k} - \boldsymbol{\beta}^*_{\leq k}\right\|^2_{\boldsymbol{\Sigma}_{\leq k}} \lesssim \frac{\frac{1}{|S_0|^2}\left\|\boldsymbol{\xi}^\top_{S_0}\widetilde{\boldsymbol{X}}_{S_0,\leq k}\right\|^2_2}{\left(1 - \frac{1}{|S_0|}\left\|\widetilde{\boldsymbol{X}}^\top_{S_0,\leq k}\widetilde{\boldsymbol{X}}_{S_0,\leq k} - |S_0|\boldsymbol{I}\right\|_2\right)^2} + \frac{\frac{1}{|S_0|}\left(\|\boldsymbol{\xi}\|^2_2 - \|\boldsymbol{\xi}_{S_0}\|^2_2\right)}{1 - \frac{1}{|S_0|}\left\|\widetilde{\boldsymbol{X}}^\top_{S_0,\leq k}\widetilde{\boldsymbol{X}}_{S_0,\leq k} - |S_0|\boldsymbol{I}\right\|_2}$$

Denote the event $A = \{1 - \frac{1}{|S_0|}\left\|\widetilde{\boldsymbol{X}}^\top_{S_0,\leq k}\widetilde{\boldsymbol{X}}_{S_0,\leq k} - |S_0|\boldsymbol{I}\right\|_2 \geq \Theta(1)\}$. We know from Lemma C.2 that $\mathbb{P}(A) \geq 1 - 1/\mathrm{poly}(d)$.

Taking expectation over $X_{S_0,\leq k}$ on both side conditional on $A$, using Lemma C.4, $0 \leq n - |S_0| = |S_1| \lesssim np_{>k}$ from Lemma A.1 and standard Gaussian concentration on $\boldsymbol{\xi}$ we have

$$\mathbb{E}_{\boldsymbol{X}_{S_0,\leq k}}\left[\left\|\hat{\boldsymbol{\beta}}_{\leq k} - \boldsymbol{\beta}^*_{\leq k}\right\|^2_{\boldsymbol{\Sigma}_{\leq k}} \Big| A\right] \lesssim \sigma^2\left(\frac{k}{n} + p_{>k}\right).$$

On the event that $A$ does not happen, it is easy to see there is a (very loose) bound on $\left\|\hat{\boldsymbol{\beta}}_{\leq k}\right\|_2 \leq d$ since the total training loss is at most $\sigma^2 n$. Therefore,

$$\mathbb{E}_{\boldsymbol{X}_{S_0,\leq k}}\left[\left\|\hat{\boldsymbol{\beta}}_{\leq k} - \boldsymbol{\beta}^*_{\leq k}\right\|^2_{\boldsymbol{\Sigma}_{\leq k}}\right]$$

$$\leq \mathbb{E}_{\boldsymbol{X}_{S_0,\leq k}}\left[\left\|\hat{\boldsymbol{\beta}}_{\leq k} - \boldsymbol{\beta}^*_{\leq k}\right\|^2_{\boldsymbol{\Sigma}_{\leq k}} \Big| A\right]\mathbb{P}(A) + \mathbb{E}_{\boldsymbol{X}_{S_0,\leq k}}\left[\left\|\hat{\boldsymbol{\beta}}_{\leq k} - \boldsymbol{\beta}^*_{\leq k}\right\|^2_{\boldsymbol{\Sigma}_{\leq k}} \Big| A^c\right](1 - \mathbb{P}(A)) \lesssim \sigma^2\left(\frac{k}{n} + p_{>k}\right).$$

This also implies $\mathbb{E}_{\boldsymbol{X}_{\leq k}}\left[\left\|\hat{\boldsymbol{\beta}}_{\leq k} - \boldsymbol{\beta}^*_{\leq k}\right\|^2_{\boldsymbol{\Sigma}_{\leq k}}\right] \lesssim \sigma^2(\frac{k}{n} + p_{>k})$.

**item 2** First from Lemma B.1 we know all data have either 0 or 1 long tail feature $[n] = S_0 \cup S_1$. We can further decompose $S_1$ into $S_1 = \cup_{i\in\mathcal{F}_1\setminus\mathcal{F}_0}S_{1,i}$, where $S_{1,i}$ represents the set of data that contains only common features and $i$-th long tail feature (that is $(\boldsymbol{x}_j)_i \neq 0$ for $j \in S_{1,i}$).

For $i$-th long tail features, we only focus on those show up in $S_{1,i}$. These data are the only data that depends on $i$-th feature. Thus, since $\hat{\boldsymbol{\beta}}$ is the minimizer of training loss, we must choose $\hat{\beta}_i$ to minimize these data in $S_{1,i}$ (assuming other $\hat{\beta}_j$ are fixed). Thus, choosing $\hat{\beta}_i = \beta^*_i$ gives a simple training loss bound on these data:

$$\left\|\boldsymbol{X}_{S_{1,i},\{\leq k,i\}}\hat{\boldsymbol{\beta}}_{\{\leq k,i\}} - \boldsymbol{y}_{S_{1,i}}\right\|^2_2 \leq \left\|\boldsymbol{X}_{S_{1,i},\leq k}(\hat{\boldsymbol{\beta}}_{\leq k} - \boldsymbol{\beta}^*_{\leq k}) - \boldsymbol{\xi}_{S_{1,i}}\right\|^2_2$$

To simplify the notation, we will drop the subscript $S_{1,i}$ in the below 2 equations.

Note that

$$\left\|\boldsymbol{X}_{\{\leq k,i\}}\hat{\boldsymbol{\beta}}_{\{\leq k,i\}} - \boldsymbol{y}\right\|_2 = \left\|\boldsymbol{X}_{\leq k}(\hat{\boldsymbol{\beta}}_{\leq k} - \boldsymbol{\beta}^*_{\leq k}) + \boldsymbol{X}_i(\beta_i - \beta^*_i) - \boldsymbol{\xi}\right\|_2$$

$$\geq \left\|\boldsymbol{X}_i(\hat{\beta}_i - \beta^*_i)\right\|_2 - \left\|\boldsymbol{X}_{\leq k}(\hat{\boldsymbol{\beta}}_{\leq k} - \boldsymbol{\beta}^*_{\leq k})\right\|_2 - \|\boldsymbol{\xi}\|_2$$

$$= \sqrt{|S_{1,i}|}|\hat{\beta}_i - \beta^*_i| - \left\|\boldsymbol{X}_{\leq k}(\hat{\boldsymbol{\beta}}_{\leq k} - \boldsymbol{\beta}^*_{\leq k})\right\|_2 - \|\boldsymbol{\xi}\|_2.$$

Thus, we have

$$\sqrt{|S_{1,i}|}|\hat{\beta}_i - \beta^*_i| \leq 2\left\|\boldsymbol{X}_{\leq k}(\hat{\boldsymbol{\beta}}_{\leq k} - \boldsymbol{\beta}^*_{\leq k})\right\|_2 + 2\|\boldsymbol{\xi}\|_2 \leq 2\left\|\boldsymbol{X}_{\leq k}\boldsymbol{\Sigma}^{-1/2}_{\leq k}\right\|_2\left\|\hat{\boldsymbol{\beta}}_{\leq k} - \boldsymbol{\beta}^*_{\leq k}\right\|_{\boldsymbol{\Sigma}_{\leq k}} + 2\|\boldsymbol{\xi}\|_2.$$

Rearranging terms and taking expectation over both side (over data generating process on common part $\boldsymbol{X}_{\leq k}$) we have

$$\mathbb{E}_{\boldsymbol{X}_{\leq k}}\left[|\hat{\beta}_i - \beta^*_i|^2\right] \lesssim \mathbb{E}_{\boldsymbol{X}_{\leq k}}\left[\frac{1}{|S_{1,i}|}\left\|\boldsymbol{X}_{S_{1,i},\leq k}\boldsymbol{\Sigma}^{-1/2}_{\leq k}\right\|^2_2\mathbb{E}_{\boldsymbol{X}_{S_0,\leq k}}\left[\left\|\hat{\boldsymbol{\beta}}_{\leq k} - \boldsymbol{\beta}^*_{\leq k}\right\|^2_{\boldsymbol{\Sigma}_{\leq k}}\right]\right] + \frac{1}{|S_{1,i}|}\|\boldsymbol{\xi}_{S_{1,i}}\|^2_2$$

$$\lesssim k\sigma^2\left(\frac{k\ln^2 d}{n} + p_{>k}\right) + \sigma^2\ln d,$$

where we use $\mathbb{E}\left[\frac{1}{|S_{1,i}|}\left\|\boldsymbol{X}_{\leq k}\boldsymbol{\Sigma}^{-1/2}_{\leq k}\right\|^2_2 \Big| |S_{1,i}|\right] \leq \mathbb{E}\left[\frac{1}{|S_{1,i}|}\left\|\boldsymbol{X}_{\leq k}\boldsymbol{\Sigma}^{-1/2}_{\leq k}\right\|^2_F \Big| |S_{1,i}|\right] = k$, item 1 and standard Gaussian concentration on $\boldsymbol{\xi}$.

**item 3** From the discussion at the beginning of the proof, we know $|S_{\geq 2}| = 0$, that is these feature does not show up in the data. Thus, we know the result is directly implied by the meaning of min-$\ell_2$-norm solution. $\qquad\square$

## C  Technical lemma

We collect few technical lemma that are used in the proof.

**Lemma C.1.** *If $p_i \leq 1$ for $i > k$, then*

$$\prod_{i>k}(1 - p_i) \geq 1 - p_{>k}.$$

*Proof.* We have

$$\prod_{i>k}(1 - p_i) = 1 - p_{>k} + \sum_{j=2}^{d} \underbrace{\sum_{k<i_1<i_2<\cdots<i_j\leq d}(-1)^j p_{i_1}\cdots p_{i_j}}_{R_j}.$$

When $j$ is even, first note that $R_j > 0$. Also,

$$
\begin{aligned}
R_j + R_{j+1} &= \sum_{k<i_1<i_2<\cdots<i_j\leq d}(-1)^j p_{i_1}\cdots p_{i_j} + \sum_{k<i_1<i_2<\cdots<i_{j+1}\leq d}(-1)^{j+1} p_{i_1}\cdots p_{i_{j+1}} \\
&= \sum_{k<i_1<i_2<\cdots<i_j\leq d}(-1)^j p_{i_1}\cdots p_{i_j}\Big(1 - \sum_{i_j<i_{j+1}\leq d} p_{i_{j+1}}\Big) \\
&\geq 0,
\end{aligned}
$$

as long as $p_{>k} \leq 1$.

Therefore, we know $\sum_{j=2}^{d} R_j \geq 0$ when $p_{>k} \leq 1$. This implies

$$\prod_{i>k}(1 - p_i) \geq 1 - p_{>k}.$$

When $p_{>k} > 1$, since $p_i < 1$ for $i > k$, so

$$\prod_{i>k}(1 - p_i) \geq 0 \geq 1 - p_{>k}.$$

$\qquad\square$

The lemma below shows the data matrix $\boldsymbol{X}_{\leq k}^{\top}\boldsymbol{X}_{\leq k}$ after normalization is close to identity.

**Lemma C.2.** *Given $\boldsymbol{X} \in \mathbb{R}^{N\times d}$ following the data generating process* (1) *with $N$ data, for $k$ that satisfies $kp_k \lesssim 1$, we have with probability at least $1 - 1/\mathrm{poly}(d)$*

$$\left\|\boldsymbol{\Sigma}_{\leq k}^{-1/2}\boldsymbol{X}_{\leq k}^{\top}\boldsymbol{X}_{\leq k}\boldsymbol{\Sigma}_{\leq k}^{-1/2} - N\boldsymbol{I}\right\|_2 \leq \Theta\left(N\left(\sqrt{\frac{\ln d}{Np_k}} + \frac{\ln^2 d}{Np_k}\right)\right)$$

*When $Np_k \gtrsim \ln^2 d$, we have $\boldsymbol{X}_{\leq k}^{\top}\boldsymbol{X}_{\leq k}$ is rank $k$. Same bound also hold for expectation.*

*Proof.* We are going to use matrix Bernstein inequality (Lemma C.5). Denote normalized matrix $\widetilde{\boldsymbol{X}}_{\leq k} = \sqrt{p_k}\boldsymbol{X}_{\leq k}\boldsymbol{\Sigma}_{\leq k}^{-1/2}$.

For $\widetilde{\boldsymbol{X}}_{\leq k}^{\top}\widetilde{\boldsymbol{X}}_{\leq k}$, to use matrix Bernstein inequality (Lemma C.5), note that

$$\widetilde{\boldsymbol{X}}_{\leq k}^{\top}\widetilde{\boldsymbol{X}}_{\leq k} = p_k\sum_{i=1}^{N}\widetilde{\boldsymbol{x}}_{i,\leq k}\widetilde{\boldsymbol{x}}_{i,\leq k}^{\top}$$

where $\widetilde{\boldsymbol{x}}_{\leq k} = \boldsymbol{\Sigma}_{\leq k}^{-1/2}\boldsymbol{x}_{\leq k}$ and for all $i \in [N]$

$$\mathbb{E}[\widetilde{\boldsymbol{x}}_{i,\leq k}\widetilde{\boldsymbol{x}}_{i,\leq k}^\top] = \boldsymbol{I}_k,$$

$$\left\|\widetilde{\boldsymbol{x}}_{i,\leq k}\widetilde{\boldsymbol{x}}_{i,\leq k}^\top - \mathbb{E}[\widetilde{\boldsymbol{x}}_{i,\leq k}\widetilde{\boldsymbol{x}}_{i,\leq k}^\top]\right\|_2 = \left\|\widetilde{\boldsymbol{x}}_{i,\leq k}\widetilde{\boldsymbol{x}}_{i,\leq k}^\top - \boldsymbol{I}_k\right\|_2 \leq k + \Theta\left(\frac{\ln d}{p_k}\right) =: L,$$

where we use Lemma C.3. We also have

$$\left\|\sum_i \mathbb{E}\left[(\widetilde{\boldsymbol{x}}_{i,\leq k}\widetilde{\boldsymbol{x}}_{i,\leq k}^\top - \mathbb{E}[\widetilde{\boldsymbol{x}}_{i,\leq k}\widetilde{\boldsymbol{x}}_{i,\leq k}^\top])(\widetilde{\boldsymbol{x}}_{i,\leq k}\widetilde{\boldsymbol{x}}_{i,\leq k}^\top - \mathbb{E}[\widetilde{\boldsymbol{x}}_{i,\leq k}\widetilde{\boldsymbol{x}}_{i,\leq k}^\top])^\top]\right\|_2$$

$$= N\left\|\mathbb{E}[\|\widetilde{\boldsymbol{x}}_{\leq k}\|_2^2 \widetilde{\boldsymbol{x}}_{\leq k}\widetilde{\boldsymbol{x}}_{\leq k}^\top] - (\mathbb{E}[\widetilde{\boldsymbol{x}}_{\leq k}\widetilde{\boldsymbol{x}}_{\leq k}^\top])^2\right\|_2$$

$$= N\left\|\mathrm{diag}(p_1^{-1} + k - 1, \ldots, p_k^{-1} + k - 1)\right\|_2$$

$$= N(p_k^{-1} + k - 1)$$

Thus, by matrix Bernstein inequality (Lemma C.5), we have

$$\mathbb{P}(\left\|\widetilde{\boldsymbol{X}}_{\leq k}^\top\widetilde{\boldsymbol{X}}_{\leq k} - \mathbb{E}[\widetilde{\boldsymbol{X}}_{\leq k}^\top\widetilde{\boldsymbol{X}}_{\leq k}]\right\|_2 \geq t) \leq 2k \exp\left(\frac{-t^2/2}{Np_k(1 + (k-1)p_k) + tLp_k/3)}\right)$$

Taking $t = \Theta(\sqrt{Np_k \ln d} + \ln^2 d) \geq \Theta(\sqrt{Np_k(1 + (k-1)p_k) \ln d} + Lp_k \ln d)$, we get

$$\mathbb{P}\left(\left\|\widetilde{\boldsymbol{X}}_{\leq k}^\top\widetilde{\boldsymbol{X}}_{\leq k} - \mathbb{E}[\widetilde{\boldsymbol{X}}_{\leq k}^\top\widetilde{\boldsymbol{X}}_{\leq k}]\right\|_2 \geq \Theta\left(\sqrt{Np_k \ln d} + \ln^2 d\right)\right) \leq 1/\mathrm{poly}(d).$$

Since $\mathbb{E}[\widetilde{\boldsymbol{X}}_{\leq k}^\top\widetilde{\boldsymbol{X}}_{\leq k}] = np_k\boldsymbol{I}_k$ and $\widetilde{\boldsymbol{X}}_{\leq k} = \sqrt{p_k}\boldsymbol{X}_{\leq k}\boldsymbol{\Sigma}_{\leq k}^{-1/2}$, we know with probability $1 - 1/\mathrm{poly}(d)$

$$\left\|\boldsymbol{\Sigma}_{\leq k}^{-1/2}\boldsymbol{X}_{\leq k}^\top\boldsymbol{X}_{\leq k}\boldsymbol{\Sigma}_{\leq k}^{-1/2} - N\boldsymbol{I}\right\|_2 \leq \Theta\left(\sqrt{\frac{N \ln d}{p_k}} + \frac{\ln^2 d}{p_k}\right)$$

and $\boldsymbol{X}_{\leq k}^\top\boldsymbol{X}_{\leq k}$ is rank $k$ when $Np_k \gtrsim \ln^2 d$. $\qquad\square$

Similar bounds can also be shown for normalized $\widetilde{\boldsymbol{x}}_{\leq k}$ and $\widetilde{\boldsymbol{x}}_{>k}$.

**Lemma C.3.** *There exists an universal large enough constant $c$ such that with probability at least $1 - 1/\mathrm{poly}(d)$, we have*

$$\|\widetilde{\boldsymbol{x}}_{\leq k}\|_2^2 \leq k + c\left(\sqrt{\sum_{i \leq k} p_i^{-1}(1 - p_i)\ln d} + p_k^{-1}\ln d\right).$$

*As a corollary, with probability at least $1 - 1/\mathrm{poly}(d)$ we have for all $i \in [n]$*

$$\|\widetilde{\boldsymbol{x}}_{i,\leq k}\|_2^2 \leq k + c\left(\sqrt{\sum_{i \leq k} p_i^{-1}(1 - p_i)\ln d} + p_k^{-1}\ln d\right).$$

*If $kp_k \lesssim 1$, the above bound can be simplify to $k + \frac{2c\ln d}{p_k}$.*

*Proof.* According to our data generation process (2), we know for data $\boldsymbol{x}$ each coordinate $x_\ell^2$ follows Bernoulli with $p_\ell$. Thus, each coordinate $\widetilde{\boldsymbol{x}}_\ell^2 \sim p_\ell^{-1}\mathrm{Bern}(p_\ell)$ for $\ell \leq k$.

Therefore, by Bernstein's inequality (Lemma C.5), we know

$$\mathbb{P}(\|\widetilde{\boldsymbol{x}}_{\leq k}\|_2^2 > \mathbb{E}[\|\widetilde{\boldsymbol{x}}_{\leq k}\|_2^2] + t) \leq \exp\left(-\frac{t^2/2}{\sum_{i \leq k} p_i^{-1}(1 - p_i) + tp_k^{-1}/6}\right)$$

Taking $t = \Theta\left(\sqrt{\sum_{i \leq k} p_i^{-1}(1 - p_i)\ln d} + p_k^{-1}\ln d\right)$ with large enough hidden constant $c$, we have

$$\mathbb{P}\left(\|\widetilde{\boldsymbol{x}}_{\leq k}\|_2^2 \geq k + \Theta\left(\sqrt{\sum_{i \leq k} p_i^{-1}(1 - p_i)\ln d} + p_k^{-1}\ln d\right)\right) \leq 1/\mathrm{poly}(d)$$

By union bound, we know for all $i \in [n]$, the same bound hold for all $\|\widetilde{\boldsymbol{x}}_{i,\leq k}\|_2$. $\qquad\square$

**Lemma C.4.** *We have*

$$\mathbb{E}_{\boldsymbol{X}_{\leq k}}\left[\left\|\boldsymbol{\Sigma}_{\leq k}^{-1/2}\boldsymbol{X}_{\leq k}^{\top}\boldsymbol{\xi}\right\|_2\right] \leq \sqrt{k}\,\|\boldsymbol{\xi}\|_2$$

*Proof.* We have

$$\left(\mathbb{E}_{\boldsymbol{X}_{\leq k}}\left[\left\|\boldsymbol{\Sigma}_{\leq k}^{-1/2}\boldsymbol{X}_{\leq k}^{\top}\boldsymbol{\xi}\right\|_2\right]\right)^2 \leq \mathbb{E}_{\boldsymbol{X}_{\leq k}}\left[\left\|\boldsymbol{\Sigma}_{\leq k}^{-1/2}\boldsymbol{X}_{\leq k}^{\top}\boldsymbol{\xi}\right\|_2^2\right] = k\,\|\boldsymbol{\xi}\|_2^2.$$

$\square$

## C.1 MATRIX CONCENTRATION INEQUALITY

We collect few standard Matrix concentration inequalities here. The first one is the standard matrix Bernstein inequality.

**Lemma C.5** (Matrix Bernstein Inequality (Corollary 6.1.2 in Tropp et al. (2015))). *Consider a finite sequence $\{\boldsymbol{S}_k\}$ of independent random matrices with common dimension $d_1 \times d_2$. Assume that each matrix has uniformly bounded deviation from its mean:*

$$\|\boldsymbol{S}_k - \mathbb{E}[\boldsymbol{S}_k]\|_2 \leq L \quad \text{for each index } k.$$

*Introduce the sum*

$$\boldsymbol{Z} = \sum_k \boldsymbol{S}_k,$$

*and let $\nu(\boldsymbol{Z})$ denote the matrix variance statistic for $\boldsymbol{Z}$:*

$$\nu(\boldsymbol{Z}) = \max\left\{\left\|\mathbb{E}\big[(\boldsymbol{Z} - \mathbb{E}[\boldsymbol{Z}])(\boldsymbol{Z} - \mathbb{E}[\boldsymbol{Z}])^{\top}\big]\right\|_2, \; \left\|\mathbb{E}\big[(\boldsymbol{Z} - \mathbb{E}[\boldsymbol{Z}])^{\top}(\boldsymbol{Z} - \mathbb{E}[\boldsymbol{Z}])\big]\right\|_2\right\}$$

$$= \max\left\{\left\|\sum_k \mathbb{E}\big[(\boldsymbol{S}_k - \mathbb{E}[\boldsymbol{S}_k])(\boldsymbol{S}_k - \mathbb{E}[\boldsymbol{S}_k])^{\top}\big]\right\|_2, \; \left\|\sum_k \mathbb{E}\big[(\boldsymbol{S}_k - \mathbb{E}[\boldsymbol{S}_k])^{\top}(\boldsymbol{S}_k - \mathbb{E}[\boldsymbol{S}_k])\big]\right\|_2\right\}.$$

*Then*

$$\mathbb{E}\,\|\boldsymbol{Z} - \mathbb{E}[\boldsymbol{Z}]\|_2 \leq \sqrt{2\nu(\boldsymbol{Z})\ln\big(d_1 + d_2\big)} + \frac{1}{3}L\ln\big(d_1 + d_2\big).$$

*In particular, for all $t \geq 0$,*

$$\mathbb{P}\big\{\|\boldsymbol{Z} - \mathbb{E}[\boldsymbol{Z}]\|_2 \geq t\big\} \leq (d_1 + d_2)\exp\Big(\frac{-t^2/2}{\nu(\boldsymbol{Z}) + Lt/3}\Big).$$

Below is the standard Chernoff's bound for Bernoulli random variables.

**Lemma C.6** (Chernoff Bounds). *Let $X = \sum_{i=1}^n X_i$, where $X_i = 1$ with probability $p_i$, and $X_i = 0$ with probability $1 - p_i$, and all $X_i$ are independent. Let $\mu = \mathbb{E}(X) = \sum_{i=1}^n p_i$. Then*

$$\mathbb{P}(|X - \mu| \geq \delta\mu) \leq e^{-\mu\delta^2/3} \quad \text{for all } 0 < \delta < 1.$$

## D EXPERIMENT DETAILS AND ADDITIONAL EXPERIMENTS

In this section, we provide details of the experiments discussed in Section 5, along with a few additional experiments.

- In Appendix D.1, we describe experiments on a synthetic toy dataset.
- In Appendix D.2, we present details for Section 5.2, which explores a modified version of MNIST with structured label imbalance, along with new results using a different target function.
- In Appendix D.3, we provide details for Section 5.3, where we augment an MNIST-based dataset with rarer samples from Omniglot and include an additional experiment.
- In Appendix D.4, we report the compute we use for the experiments.

- In Appendix D.5, we show that long-tail data exhibit high memorization scores when measured by influence functions (Feldman & Zhang, 2020).

For experiments in this section, all datasets contain original labels ranging from 0 to 9. We deliberately control the number of samples per label in the training set to induce a long-tail distribution. More precisely, the number of samples for labels $i \in \{0, 1, \ldots, 9\}$ follows Zipf's law with exponent $s = 2$. The unnormalized weight assigned to rank $i$ is defined as $w_i = (i + 1)^{-s}$ and the final sampling probability for label $i$ is $p_i = w_i / \sum_{j=0}^{9} w_j$.

Without loss of generality, we primarily focus on the samples with label **9** as the representative long-tail feature in the following experiments except for the experiment described in Appendix D.3. We further analyze the performance of models that successfully memorize all long-tail features, with a particular emphasis on their generalization behavior on the test dataset.

To analyze the compositional effect, we focus on a particular long-tail feature and vary its frequency of appearance in the training set across 10 levels, ranging from 100 to 550 with increments of 50. For each occurrence level, we train $n$ independently initialized models and evaluate their performance on the test set. Specifically, we report the average test loss $\mu_{\text{test\_loss}}$ and the variance of this average $\text{Var}(\bar{x}) = \sigma^2/n$.

### D.1 Additional Synthetics Experiment: Gaussian mixtures

**Dataset Construction**   We generate synthetic data based on multivariate normal distributions. Specifically, we randomly initialize 10 mean vectors, each corresponding to a digit from 0 to 9. Each mean vector is of dimension 100 and is sampled from a multivariate normal distribution with a mean of a zero vector and an identity covariance matrix. For each class $c \in \{0, 1, ..., 9\}$, samples are drawn from a multivariate normal distribution with the corresponding mean vector and a covariance matrix of $\sigma I$. The number of samples per class follows Zipf's law decay distribution.

To construct the final dataset, we randomly sample triplets from the generated data, where each triplet consists of three individual samples. The label of a triplet is defined as the sum of the labels of its constituent samples. This results in a dataset of shape $n \times 300$.

**Model Architecture**   We employ a two-layer neural network with an additive structure. During the forward pass, the model processes the input by dividing the 300-dimensional feature vector into three segments: dimensions 0–100, 100–200, and 200–300. The network independently computes the outputs for each segment, and the final prediction is obtained by summing the three outputs.

**Training Setup**   We train the model using Mean Squared Error (MSE) loss and optimize it using the Adam optimizer. We use a learning rate of **0.0001**, train for **300** epochs, with a batch size of **256**.

**Evaluation Metric**   To assess the compositional effect of memorizing long-tailed features on generalization, we visualize the results using a performance plot. The x-axis represents the total number of distinct '9's the model has encountered during training, while the y-axis shows the corresponding test loss. For each x-axis value (ranging from 100 to 550), we train 20 independent models. Each data point is accompanied by an error bar, indicating the variance of mean of loss across test samples—smaller variance suggests higher confidence in the loss estimates.

Our test dataset is constructed as follows: we first generate 600 vectors for each class (0 through 9) by sampling from the same class-wise mean vectors used in the training dataset. To evaluate out-of-distribution generalization on the long-tail feature, we focus on class 9. For each vector labeled as 9, we randomly pair it with vectors from all other classes to create 2700 test samples. This results in a total of $600 \times 2700 = 1,620,000$ test examples. The goal of this setup is to examine whether memorization of the long-tail feature (class 9) generalizes to unseen vectors with the same label. If the model exhibits improved performance on these test examples, it suggests that long-tail memorization contributes positively to generalization.

As shown in Figure 3, the test loss on examples involving class 9 decreases as the number of class 9 samples in the training set increases. Notably, the class 9 vectors in the test set are not present in the training set, indicating that the observed performance gain reflects genuine generalization rather than rote memorization.

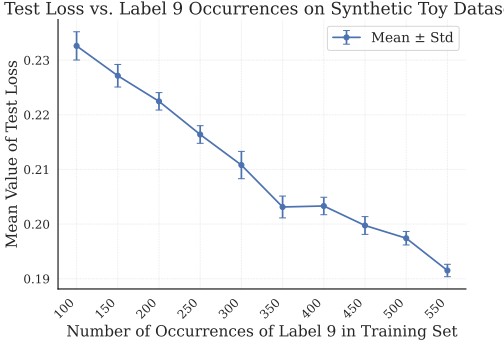

Figure 3: Test loss as a function of the number of training occurrences of class 9.

## D.2 EXPERIMENT DETAILS IN SECTION 5.2: LONG-TAIL MNIST DATA

We give more details of the experiments in Section 5.2.

**Dataset Construction**  In the training set, the number of samples per label is as follows: {0: 77,435; 1: 19,358; 2: 8,603; 3: 4,839; 4: 3,097; 5: 2,150; 6: 1,580; 7: 1,209; 8: 955; 9: 774}, resulting in a total of 120,000 samples.

It is worth noting that the original MNIST training set contains roughly 5,000–6,000 samples per digit. Therefore, for labels 0–3, we apply *oversampling* to increase their frequency. All samples are then randomly shuffled and grouped into triplets, yielding a final training set of 40,000 triplets.

For the test set, we uniformly sample 200 samples for each class label from 0 to 4. Additionally, we sample 200 unseen digit-9 images (i.e., samples not included in the training set). Similar to the construction in the toy setting, test samples are generated by pairing one digit-9 sample with each of the $(200 \times 5)$ combinations from labels 0–4. In total, this results in $200 \times 200 \times 5/2 = 100,000$ test samples.

We clarify that all training samples are drawn exclusively from the MNIST *training* set, and all test samples are drawn from the MNIST *test* set.

**Training Setup**  We train the model using Mean Squared Error (MSE) loss and optimize it using the Adam optimizer. We use a learning rate of **0.0001**, train for **300** epochs, with a batch size of **256**, and no weight decay.

**Evaluation Metric**  To evaluate how memorization of long-tailed features influences compositional generalization in a real-world setting, we adopt the same performance visualization approach as in our toy experiments.

For each x-axis value (ranging from 100 to 550), similarly, we train 100 independent models. Each model is trained on an 80% random subset of the original dataset, ensuring that every subset contains the same number of '9's. The reported test loss is the average over these 100 models.

Notably, since we control the number of '9's in the training subsets, models trained with more '9's are exposed to fewer instances of other digits. However, this does not significantly impact the observed downward trend in test loss. This further supports our hypothesis that memorizing long-tailed features substantially aids generalization.

### D.2.1 ADDITIONAL EXPERIMENT: ROBUSTNESS ACROSS TARGET FUNCTIONS

We further examine the robustness of the observed trends by modifying the target function used during training. Specifically, instead of using a simple sum of three labels as in Section 5.2, we redefine the target as a weighted combination:

$$label = \sum_{i=1}^{3} i \times label_i$$

This target is slightly more complex than the one in Section 5.2.

In Figure 4, we observe that the loss curves continue to exhibit a similar downward trend as the number of training instances from class 9 increases. This indicates our results in Section 5.2 is robust to different target.

Notably, the dataset and training setup are exactly the same as in the previous experiments. The only difference lies in the way labels are computed in the dataset.

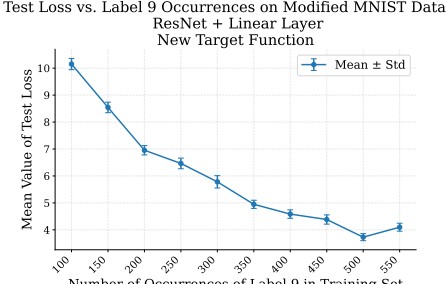 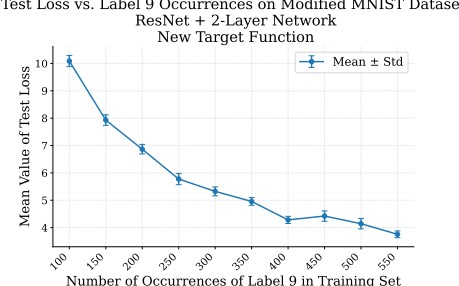

Figure 4: **(Left)** ResNet+**Linear layer** with New Target Function: Test loss as a function of the number of training occurrences of class 9; **(Right)** ResNet+**2-layer network** with New Target Function: Test loss as a function of the number of training occurrences of class 9.

### D.3 EXPERIMENT DETAILS IN SECTION 5.3: RARER SAMPLE CASE FROM OMNIGLOT DATASET

Firstly, we give more details of the experiments in Section 5.3.

**Dataset Construction**  The MNIST part of the mixed dataset is constructed in the same way as the training set described in Appendix D.2. The only difference is that we additionally include samples from the Omniglot dataset. Specifically, we randomly select one sample from each class labeled 0–9 in the Omniglot training set, resulting in 10 Omniglot samples. These samples are preprocessed to match the format and size of the MNIST data and then mixed into the MNIST training set. To ensure the total number of samples is divisible by 3 (since samples are grouped into triplets), we randomly remove one MNIST sample after adding the Omniglot data. The final training set size is 40,003.

For the test set, we uniformly sample 100 MNIST samples from each class label 0–4, resulting in 500 MNIST test samples. We then consider all possible pairwise combinations among the 10 Omniglot samples, generating 45 unique Omniglot pairs. Each of these pairs is combined with the MNIST samples to form test triplets, resulting in a total of $45 \times 500 = 22,500$ test samples.

**Training Setup**  We also train the model using Mean Squared Error (MSE) loss and optimize it using the Adam optimizer. We use a learning rate of **0.0001**, train for **300** epochs, with a batch size of **256**.

### D.3.1 ADDITIONAL EXPERIMENT: MORE OMNIGLOT DATA PER CLASS

Similar as Section 5.3, we also design experiments where Omniglot samples appear more frequently, and the test sample contains only one instance from Omniglot. Specifically, we introduce a variant of the 3-digit synthetic dataset by injecting distractor samples from the Omniglot dataset. To construct the training set, we randomly select 10 samples for each digit label (0–9) from Omniglot (totaling 100 samples) and mix them into the original training data. For the test set, we sample the same 10 Omniglot samples in training dataset per digit label (0–9). Additionally, we sample 200 test digits each from MNIST labels 0–4.

Each final test instance is formed by combining one digit from Omniglot and two digits from MNIST, resulting in test samples that include both known and out-of-distribution (OOD) components. In total, this produces $100 \times 5 \times 100 = 50,000$ test samples.

Table 2 show that the model still performs well even on test data that are not limited to long-tail feature combinations.

Table 2: Stronger Memorization Behavior of ResNet Variants Under Different Weight Decay Settings

| Model | Memorization (WD = 0) | No Memorization (WD = 0.5) |
|---|---|---|
| Sum | Test: 0.0697 
 Train: 0.0002 | Test: 0.8266 
 Train: 0.0538 |
| Linear | Test: 0.1171 
 Train: 0.0006 | Test: 0.9571 
 Train: 0.0715 |
| 2-Layer | Test: 0.0650 
 Train: 0.0013 | Test: 0.6579 
 Train: 0.0196 |

## D.4 COMPUTE REQUIREMENTS

The computational resources used for all our experiments are as follows: 10 GTX 2080 Ti GPUs with 60 GB of RAM. For the Experiments on Linear Model, completing all runs takes approximately 3 hours. For the Simple Task with Composition experiment, training a single model takes about 1 hour. For the Effect of Memorization experiment, training a single model also takes around 1 hour.

## D.5 MEMORIZATION SCORE VIA INFLUENCE FUNCTION

Although throughout this paper we directly use the training loss to measure memorization, an alternative approach is to estimate memorization using influence functions, as proposed in Feldman & Zhang (2020). In this section, we show that long-tail data indeed exhibit high memorization scores when measured by influence functions.

We implemented the influence estimation method (Feldman & Zhang, 2020) to quantify how training examples affect model performance. Since our task is a regression problem that predicts digit sums (rather than a classification task), we adapted their influence score by replacing the accuracy difference with a loss difference. Specifically, the memorization score is computed as:

$$\text{mem}(i) = \mathbb{E}[\text{loss}(x_i) \mid \text{data } i \notin \text{training set}] - \mathbb{E}[\text{loss}(x_i) \mid \text{data } i \in \text{training set}].$$

A positive memorization score indicates that including the sample in training reduces its loss, suggesting a stronger memorization of the sample.

**Experimental Setup:** We trained 100 models on randomly sampled subsets (80% of the full training set, i.e., 32,000 out of 40,000 samples) and evaluated the memorization scores for all instances of the long-tail digit 9. The setup is the same as in Section 5.2.

Table 3: Top-10 most memorized digit-9 examples. All highly memorized examples appear exactly once in the full training set.

| Original Index Containing '9' | Memorization Score |
|---|---|
| 4362 | 80.38 |
| 21445 | 80.28 |
| 9390 | 61.33 |
| 21302 | 55.62 |
| 19494 | 55.50 |
| 29286 | 49.01 |
| 31456 | 38.85 |
| 7620 | 35.90 |
| 19412 | 33.68 |
| 8847 | 32.67 |

The result is shown in Table 3. Our analysis shows substantial memorization of long-tail examples, with an overall mean memorization score of **2.41** (std 7.54) and a maximum of **80.38**. These values are much larger than our typical training loss, indicating that whether a specific long-tail example appears in the training set has a strong effect on OOD generalization.

