# OpenReview forum: "Memorizing Long-tail Data Can Help Generalization Through Composition"
_ICLR.cc/2026/Conference — ICLR 2026 Poster_

### Official Review · Reviewer_mMst · 2025-10-29

**Soundness:** 3
**Presentation:** 4
**Contribution:** 3
**Rating:** 6
**Confidence:** 4

**Summary:**

This work considers an interesting an interesting angle of how memorizing long-tail samples can be helpful for generalization. Theoretically, it proves that for linear classifier---whether the underlying distribution is noiseless or noisy---memorization can help generalization both in and out of distribution under some assumptions on the distributions. Empirically, it shows that memorization can help on an interesting construction of task: computing the sum of MNIST digits. With proper choice of model architecture, memorization can help improve the results when a certain digit is significantly under represented in the data set.

**Strengths:**

The presentation of this work is outstanding: theoretical claims are clearly defined and the underlying intuition of the proof well explained. I also appreciate the authors' effort of motivating the problem as well as presenting the related work with precise and succinct language. The theoretical claims are sound. I don't find any apparent problem in the proof, either. The design of the three-digit-sum problem is new to me. Despite some limitation of the design, which I will come to later, the idea is intriguing. Overall, this work is solid technically.

**Weaknesses:**

Compared to the outstanding presentation and rigorous theoretical formulation, this paper is slightly weaker on the potential impact. Specifically:

1) The linear case is slightly simplistic as composition is natural. If two features both contributes positively to the prediction score, i.e., have positive corresponding field in $\beta$, then observing one of them at a time in training example should suffice for good test results. In nonlinear case, e.g., an XOR, observing one feature at a time may not be sufficient for the telling the outcome when both features are present (nonzero). In fact, the sum-of-digit task is somewhat linear with respect to the digits. I wonder if the phenomenon of composition can be observed in more general tasks.

2) The notion of memorization here is slightly different from the literature I'm familiar with. I'm more used to influence score based criteria for memorization, e.g., removing a training example will significantly impact the prediction of another example. I believe this work assumes that an overparametrized model with unregularized training will memorize. Is this assumption common in literature?

3) Following 2), MNIST is a fairly 'simple' dataset for which a small size of sample can lead to good model already with or without memorization. For stronger impact, the authors may want to consider some more complex tasks.

**Questions:**

My questions are mainly on the potential impact of the work.

1) Could you provide some more real world examples of tasks where composition is natural?

2) If time allows, could you quickly check the influence score a training example on itself or on the test samples? Either a simple leave-one-out test (retrain the model with a training set differing by one entry) or the estimation in Feldman,Vitaly and Zhang.

3) What could be the future extension of the result in this work?


Feldman, Vitaly, and Chiyuan Zhang. "What neural networks memorize and why: Discovering the long tail via influence estimation." Advances in Neural Information Processing Systems 33 (2020): 2881-2891.

---

> ### Author Response · Authors · 2025-11-25
> **Author Response 1**
>
> We appreciate the reviewer’s time and effort in evaluating our work. Below, we provide detailed responses to the specific concerns raised.
>
>
> > **W1**: The linear case is slightly simplistic as composition is natural. If two features both contributes positively to the prediction score, i.e., have positive corresponding field in, then observing one of them at a time in training example should suffice for good test results. In nonlinear case, e.g., an XOR, observing one feature at a time may not be sufficient for the telling the outcome when both features are present (nonzero). In fact, the sum-of-digit task is somewhat linear with respect to the digits. I wonder if the phenomenon of composition can be observed in more general tasks.
>
> We agree that the linear model is a simplified setting and that our sum-of-digits task is indeed ''close to linear'' in its underlying digit features. This simplification is intentional: our goal is to isolate and analyze a concrete mechanism by which memorizing long-tail features can support compositional generalization, in a regime where we can prove guarantees and tightly control the data distribution.
>
> At the same time, we believe that linear/additive composition is a reasonable first-order approximation for more general settings. In practice, models operate on high-dimensional learned embeddings, and representation learning often makes downstream tasks more linear in this embedding space. A classical example is the approximate relation *King - Man + Woman $\approx$ Queen* in word embeddings [1], which suggests that certain semantic structures can be captured in an approximately linear way. Our linear analysis can be viewed as modeling this kind of ''linearized'' compositional structure at the feature level.
>
> We agree that extending beyond linear composition to explicitly nonlinear interactions (e.g., XOR-like dependencies and higher-order compositions) is an important and interesting direction. We view our results as a first step toward such extensions.
>
>
> > **W2**: The notion of memorization here is slightly different from the literature I'm familiar with. I'm more used to influence score based criteria for memorization, e.g., removing a training example will significantly impact the prediction of another example. I believe this work assumes that an overparametrized model with unregularized training will memorize. Is this assumption common in literature?
>
> In this paper we adopt the standard ''interpolation'' view of memorization used in the overparametrized learning literature: by memorization we specifically mean that the model achieves (near-)zero training loss on the training set, rather than an influence-based notion. This perspective appears in works such as [2,3,4], where highly overparametrized unregularized models are shown, both empirically and theoretically, to fit or interpolate their training data.
>
> This line of work also motivates the literature on benign overfitting [5,6], which studies the test performance of interpolating estimators in overparametrized regimes under various statistical assumptions. Our theoretical setup follows this tradition. Thus, the assumption that an overparametrized, unregularized model will memorize its training data is not new, but aligns with a large body of prior work.
>
> > **W3**: Following 2), MNIST is a fairly 'simple' dataset for which a small size of sample can lead to good model already with or without memorization. For stronger impact, the authors may want to consider some more complex tasks.
>
> We appreciate the reviewer's suggestion to extend our experiments to more complex tasks. We acknowledge that MNIST is a relatively simple dataset. However, our experiments demonstrate that even on this simple task, memorization remains essential. As shown in Table 1 in our paper, when we prevent memorization by applying weight decay, the test performance degrades significantly. This demonstrates that memorization is necessary for compositional generalization even in the MNIST setting.
>
> On the other hand, we agree that more complex tasks would be highly interesting to explore. Examples include reasoning and language modeling tasks in LLMs, as well as compositional zero-shot learning in vision tasks. These represent promising directions for our future work, which we indeed want to explore more.
>
> > **Q1**: Could you provide some more real world examples of tasks where composition is natural?
>
> We are happy to provide a few more concrete examples where linear or additive composition arises naturally. In general, tasks that require composing ''sub-concepts'' are good examples:
>
> - Question answering about rare entities:
>
>     A model may memorize facts about low-frequency entities or attributes (long-tail features) and then answer questions that combine several such facts in new ways.
>
> - Personalized systems:
>
>     User-specific rare events can be memorized and then combined to answer new queries or recommend more relevant content.

---

> ### Author Response · Authors · 2025-11-25
> **Author Response 2**
>
> > **Q2**: If time allows, could you quickly check the influence score a training example on itself or on the test samples? Either a simple leave-one-out test (retrain the model with a training set differing by one entry) or the estimation in Feldman,Vitaly and Zhang.
>
> We thank the reviewer for this valuable suggestion. Following the reviewer's recommendation, we implemented the influence estimation method from Feldman and Zhang [7] to quantify how training examples affect model performance. Since our task is a regression problem that predicts digit sums (rather than a classification task), we adapted their influence score by replacing the accuracy difference with a loss difference. Specifically, the memorization score is computed as:
>     $$\text{mem}(i) = \mathbb{E}[\text{loss}(x_i) \mid  \text{data $i \notin$ training set}] - \mathbb{E}[\text{loss}(x_i) \mid \text{data $i \in$ training set}].$$
>     A positive memorization score indicates that including the sample in training reduces its loss, suggesting a stronger memorization of the sample.
>
> **Experimental Setup:** We trained 100 models on randomly sampled subsets (80\% of the full training set, i.e., 32,000 out of 40,000 samples) and evaluated the memorization scores for all instances of the long-tail digit 9. The setup is the same as in Section 5.2.
>
> - **Memorization Effects:** Our analysis shows substantial memorization of long-tail examples, with an overall mean memorization score of **2.41** (std 7.54) and a maximum of **80.38** (see table). These values are much larger than our typical training loss, indicating that whether a specific long-tail example appears in the training set has a strong effect on OOD generalization. All highly memorized examples appear exactly once in the full training set.
>
> - **Support for Our Hypothesis:** These findings provide further support for our central claim: memorization of long-tail data is essential for generalization through composition. High memorization score examples show that the model’s ability to correctly handle test samples that require composition (e.g., sums involving the digit 9) depends on having memorized the corresponding long-tail training instance.
>
> | **Original Index Containing '9'** | **Avg Memorization Score** |
> |----------------------------------|-----------------------------|
> | 4362                             | 80.38                       |
> | 21445                            | 80.28                       |
> | 9390                             | 61.33                       |
> | 21302                            | 55.62                       |
> | 19494                            | 55.50                       |
> | 29286                            | 49.01                       |
> | 31456                            | 38.85                       |
> | 7620                             | 35.90                       |
> | 19412                            | 33.68                       |
> | 8847                             | 32.67                       |
>
>
> > **Q3**: What could be the future extension of the result in this work?
>
> We see several natural extensions of this work:
>
> - Theoretical extension.
>     Extend our analysis beyond linear models to nonlinear models, such as neural networks, and investigate when similar guarantees can be established.
>
> - Richer compositional structure.
>     For both theory and experiments, move beyond simple linear composition to more complex compositional patterns, such as higher-order interactions, as suggested by the reviewer.
>
> - Larger-scale empirical validation.
>     Expand from small supervised learning experiments to more complex tasks and models, including large-scale language models and more challenging vision benchmarks.
>
>
> [1] Tomas Mikolov et al., Linguistic Regularities in Continuous Space Word Representations, NAACL 2013.
>
>
> [2] Vitaly Feldman, Does learning require memorization? a short tale about a long tail, STOC 2020.
>
>
> [3] Chiyuan Zhang et al., Understanding deep learning requires rethinking generalization, ICLR 2017.
>
> [4] Devansh Arpit et al., A closer look at memorization in deep networks,
> ICML 2017.
>
> [5] Peter Bartlett et al., Benign overfitting in linear regression, PNAS 2020.
>
> [6] Trevor Hastie et al., Surprises in high dimensional ridgeless least squares interpolation, The Annals of Statistics 2022.
>
> [7] Vitaly Feldman and Chiyuan Zhang, What neural networks memorize and why: Discovering the long tail via influence estimation, NeurIPS 2020.

---

### Official Review · Reviewer_iwWd · 2025-10-31

**Soundness:** 2
**Presentation:** 2
**Contribution:** 2
**Rating:** 2
**Confidence:** 4

**Summary:**

This paper explores how memorization of rare, long-tail examples can improve generalization when combined with a model’s ability to compose known features in new ways. Through theoretical analysis in linear settings and small-scale experiments on compositional MNIST tasks, the authors show that memorization enables correct predictions on unseen combinations of rare features.

**Strengths:**

The paper provides a clear theoretical formulation connecting memorization and compositional generalization, an underexplored relationship in deep learning theory. Its synthetic and modified MNIST experiments effectively illustrate how architectural structure influences compositional ability. Finally, it contributes a valuable conceptual shift, framing memorization not purely as overfitting, but as a potentially beneficial mechanism for learning from long-tail data.

**Weaknesses:**

**Oversimplified Definition of Memorization**

The paper treats memorization as a binary property i.e., models either memorize or do not memorize. This definition ignores the nuanced ways sample level memorization actually behaves. For example, memorization scores can vary from 0 (i.e., perfect generalization) to 1 (perfect memorization). By treating the property as binary, the authors are ignoring the entire range of values between 0 and 1.


**No Empirical Verification of Memorization**

Despite repeatedly claiming that rare examples (like the digit “9”) were memorized, the authors never tested this directly. They inferred memorization from improved performance on rare-digit test cases but did not apply any established measurement techniques (e.g., Feldman et al's self influence), to verify that the model had indeed memorized those samples. Without this validation, the central claim that memorization enables composition remains speculative.

**Reliance on Indirect Behavioral Evidence**

The experimental support for memorization is limited to behavioral trends: test loss decreases as the frequency of the rare digit increases and increases when weight decay is applied. While suggestive, these results can also be explained by better statistical coverage or regularization effects rather than genuine memorization. The lack of causal evidence weakens the argument of this work.

**Questions:**

See above

---

> ### Author Response · Authors · 2025-11-25
> **Author Response 1**
>
> We appreciate the reviewer’s time and effort in evaluating our work. However, we believe the reviewer has significantly misunderstood our paper, which has led to an unfair assessment.
>
> The main confusion comes from the definition of memorization. In our paper, we follow the line of work starting from [1,3,4] and by memorization we refer specifically to achieving near-zero training loss (interpolation), not any other metric. Near-zero training loss is a standard and direct indicator of memorization in the overparametrized regime [1,3,4]. All experiments, unless otherwise specified, reach near-zero training loss and therefore memorize the training data.
>
> Below, we provide detailed responses to the specific concerns raised.
>
> > **W1**:  The paper treats memorization as a binary property i.e., models either memorize or do not memorize. This definition ignores the nuanced ways sample level memorization actually behaves. For example, memorization scores can vary from 0 (i.e., perfect generalization) to 1 (perfect memorization). By treating the property as binary, the authors are ignoring the entire range of values between 0 and 1.
>
> We respectfully disagree with this assessment. Our choice to use a seemingly ''binary'' notion is deliberate and driven by the specific research question our paper investigate.
>
> - **Our binary framing does not ignore nuance. It isolates the mechanism we study.**
>
>     We do not claim that memorization is inherently binary in all contexts. Instead, we focus on a clean and controlled question: *What happens to compositional generalization when a model can fully memorize its training set versus when it cannot?* For this purpose, the distinction between memorization and non-memorization is the relevant one. Incorporating continuous memorization scores (such as influence scores) is an interesting direction for future work, but it is orthogonal to the mechanism our paper investigates.
>
> - **Existing ''continuous'' memorization scores are also used via thresholds in practice.**
>
>     We acknowledge the value of frameworks like self-influence scores [1,2], which assign real-valued memorization scores to individual samples. However, even in that literature, analyses typically rely on thresholding these scores (for example, declaring examples with score $\ge 0.25$ as ''memorized'') and then comparing the memorized and non-memorized subsets. In other words, a binary notion remains central for interpretation. Our interpolation-based definition serves a similar purpose but is tied directly to a quantity we can control during training, namely the training loss, rather than a post-hoc diagnostic.
>
> - **Our definition of memorization provides controllability.**
>
>     While more fine-grained memorization metrics can be useful, to our knowledge there is no practical way to *set* or *target* a desired memorization score between 0 and 1 for each sample in a controlled experiment. Even in the self-influence framework, the metric is primarily *observational* rather than something one can directly manipulate.
>
>     In contrast, our near-zero training loss criterion offers a practical and interpretable way to control memorization: by adjusting weight decay, we can reliably move between regimes where the network can or cannot interpolate the data. This controllability is essential for our causal-style comparisons because it allows us to systematically enable or block memorization and then observe the resulting effect on compositional behavior.
>
> Overall, while our definition is simpler than per-example memorization scores, it is intentionally chosen to give a controllable and theoretically meaningful handle on the presence vs. absence of memorization as a mechanism for composition, which is the core focus of this work.
>
>
> > **W2**: Despite repeatedly claiming that rare examples (like the digit `9') were memorized, the authors never tested this directly. They inferred memorization from improved performance on rare-digit test cases but did not apply any established measurement techniques (e.g., Feldman et al's self influence), to verify that the model had indeed memorized those samples. Without this validation, the central claim that memorization enables composition remains speculative.
>
> We strongly disagree with this assessment. **We did not infer memorization from improved test performance on rare examples.** Instead, we enforce memorization by training models to reach near-zero training loss. As noted at the beginning of our response, by memorization we specifically mean achieving near-zero training loss, and all experiments (unless otherwise stated) are trained to reach this level. Memorization is therefore not inferred from any other signal but is directly confirmed by the near-zero training loss.

---

> ### Author Response · Authors · 2025-11-25
> **Author Response 2**
>
> > **W3**: The experimental support for memorization is limited to behavioral trends: test loss decreases as the frequency of the rare digit increases and increases when weight decay is applied. While suggestive, these results can also be explained by better statistical coverage or regularization effects rather than genuine memorization. The lack of causal evidence weakens the argument of this work.
>
> We strongly disagree with this assessment.
>
> First, as noted at the beginning of the response, by memorization we mean achieving near-zero training loss, and all experiments (unless otherwise specified) are trained to near-zero training loss. This rules out the possibility that the observed differences arise from other factors such as statistical coverage or regularization effects mentioned by the reviewer. In the relevant settings, the model has memorized the training set. **Our comparisons therefore examine regimes where memorization is present versus regimes where it is absent (for example, by strong weight decay), rather than focusing solely on behavioral trends.**
>
> Second, the experiments in Section 5.3 supports our theory that the memorization enables composition for generalization and cannot be explained by ''better statistical coverage or regularization effect''.
>
> - Statistical coverage/Sample complexity:
>
>     In the Omniglot setup, each Omniglot example appears only once in the training data, yet the model performs well on test examples that reuse that exact sample in new compositions.
>
>     **Classical statistical theory would predict improved performance with more samples, but here this mechanism is unavailable**: there is no coverage of each Omniglot sample beyond a single occurrence.
>
>     The only way for the model to succeed on these test examples is to memorize individual training instances and reuse them in new contexts, which is exactly the compositional use of memorized examples that our theory describes.
>
> - Regularization effect/Weight decay:
>
>     Using the Sum model as an example (the same argument applies to all Sum, Linear, and 2-layer models), Table 1 shows:
>
>     - Weight decay = 0: Training loss = 0.0004(memorization), test loss = 0.176
>     - Weight decay = 0.5: Training loss = 0.0245(no memorization), test loss = 2.874
>
>     **Classical statistical learning theory would suggest that regularization should improve test performance. Here, however, the opposite occurs: the model performs better when there is no regularization.** Thus the results cannot be explained by a regularization effect. Instead, without regularization the model memorizes, and this enables the compositional mechanism that leads to better performance.
>
>     Our claim is not ''regularization vs. no regularization,'' as the reviewer suggests, but **''able to memorize vs. prevented from memorizing''**. Weight decay is used as a controlled way to reduce the model’s ability to drive the training loss to near zero. When weight decay is strong enough to prevent memorization, test performance on compositional tasks degrades sharply.
>
>     This pattern is exactly what our theory predicts: when the model can memorize training instances, it can later compose these memorized pieces to solve new tasks. When strong weight decay prevents memorization, this compositional mechanism breaks down and performance deteriorates.
>
>
>
> [1] Vitaly Feldman, Does learning require memorization? a short tale about a long tail, STOC 2020.
>
> [2] Vitaly Feldman and Chiyuan Zhang, What neural networks memorize and why: Discovering the long tail via influence estimation, NeurIPS 2020.
>
> [3] Chiyuan Zhang et al., Understanding deep learning requires rethinking generalization, ICLR 2017.
>
> [4] Devansh Arpit et al., A closer look at memorization in deep networks, ICML 2017.

---

### Official Review · Reviewer_HA3C · 2025-11-08

**Soundness:** 3
**Presentation:** 3
**Contribution:** 4
**Rating:** 6
**Confidence:** 4

**Summary:**

This work builds on to the line of work by Feldman and Zhang that has studied how long tail memorization can help with generalization in deep learning.

There is a key conceptual shift that this paper makes on top of Feldman. Feldman argued that memorization helps because test examples are similar to memorized training examples, which allows the model to recall them directly. This paper adds a new dimension to this discourse: Memorization helps not only because it reproduces similar examples, but also because it enables composition. This means that combining multiple memorized rare examples can lead to generalization into new configurations.

To demonstrate this idea of compositionality, the authors move away from the singleton tasks in past work to new tasks such as (i) a “sum of three MNIST digits” setup and (ii) an MNIST–Omniglot mixture testing one-shot memorization.

The authors develop a theoretical model in which different data features follow a power-law frequency distribution.
They prove that the min norm solution which memorizes training data can correctly predict on test examples composed of multiple long-tail features that never co-occurred during training. The theoretical argument is supported via the experimental results on the newly created synthetic datasets.

Results show that networks capable of processing input components modularly (e.g., per-digit ResNets with additive aggregation) generalize compositionally, whereas architectures that entangle inputs early (“cross-channel” ResNets) fail.

The paper also shows that an attempt to mitigate memorization (such as weight decay penalty) leads to a loss in model generalizartion on such compositional tasks.

Disclosure: I have not reviewed the theory carefully.

**Strengths:**

1. Conceptual Extension: I quite like the extension this paper attempts to make over the singleton memorization argument made in Feldman et. al. The bridge is quite intellectually appealing and can connect various ideas like one-shot generalization and memorization in overparametrized models.
2. The power law based feature setup seems quite simple yet expressive. I believe this is sufficient to motivate the emprical underpinnings of the work
3. The paper has a good mix of toy tasks: from linear regression to a controlled mnist and omniglot task. I like how they are able to connect the architectural dependence here as we visualize the transition from memorizatioon to composition.

**Weaknesses:**

1. The main weakness of the work is in its experimental scope. I admit that this in general will stay a hard task but I would like to challenge the authors to find meaningful ways to extend these setups to those of more practical relevance.
     i. this requires identifying where in the real world do one-shot composition of memorized instances naturally happens
     ii. run controlled experiments to actually experiment by ablating away that capability
     iii. if the memorized composition was indeed a mechanism by which models generalize, i actually thing it is quite a useful exercise to show that this happens in real tasks. if not, why is this phenomenon of interest? i am writing this as motivation rather than actually questioning the value of this line of work, which i quite like.
2. I believe this paper also needs a discussion on when memorization hurts composition. This is especially true for scenarios such as spurious correlations. How would the theory and/or experiments inetrsect with this.

**Questions:**

1. The task of single example memorization in big models is hard. I wonder if some efforts around experimentation with PEFT, or in context examples can somehow connect here. In context learning is an example of single example generalization with high information recall (which is what you intend to use the word memorization for, anyways). This is just one thought to aid experimentation.

---

> ### Author Response · Authors · 2025-11-25
> **Author Response**
>
> We appreciate the reviewer’s time and effort in evaluating our work. Below, we provide detailed responses to the specific concerns raised.
>
> > **W1**: The main weakness of the work is in its experimental scope. I admit that this in general will stay a hard task but I would like to challenge the authors to find meaningful ways to extend these setups to those of more practical relevance. ... i am writing this as motivation rather than actually questioning the value of this line of work, which i quite like.
>
> > **Q1**: The task of single example memorization in big models is hard. I wonder if some efforts around experimentation with PEFT, or in context examples can somehow connect here. In context learning is an example of single example generalization with high information recall (which is what you intend to use the word memorization for, anyways). This is just one thought to aid experimentation.
>
> We thank the reviewer for this valuable suggestion on extending our framework to more practical settings. Exploring complex real-world scenarios is indeed an important and promising research direction.
>
> Long-tail features and compositional reasoning are pervasive characteristics in many practical tasks. As the reviewer points out, in-context learning (ICL) provides a particularly promising avenue to validate our theoretical framework. Recent work [1] has formalized ICL as a retrieval process from associative memory: pretrained models encode substantial prior knowledge (i.e., memorized instances) during training, which can be retrieved and activated at inference time using in-context examples as contextual cues.
>
> This reveals a natural connection to our framework. Our work shows that memorizing long-tail features during training enables OOD generalization through composition. ICL displays a similar mechanism at inference time: retrieval from memorized data supports compositional reasoning. More recent perspectives further interpret ICL as memory reshaping in modern Hopfield networks conditioned on in-context examples [2]. This also raises an intriguing question: whether the memorized long-tail data retrieved during ICL similarly facilitates reasoning through composition, mirroring the mechanism we analyze in our paper. Understanding this connection could provide a unified view of memorization and composition across different learning paradigms (training-time vs. inference-time).
>
> We hope our work serves as a starting point for exploring the synergy between memorization and composition in broader contexts. Our core contribution is to provide concrete theoretical and empirical examples showing when memorization can help OOD generalization through composition. We believe this perspective may help deepen our understanding of diverse learning mechanisms, including ICL, and suggests testable hypotheses about how models leverage memorized knowledge compositionally.
>
>
> > **W2**: I believe this paper also needs a discussion on when memorization hurts composition. This is especially true for scenarios such as spurious correlations. How would the theory and/or experiments inetrsect with this.
>
> We thank the reviewer for this suggestion. One reason our paper focuses on the beneficial side of memorization for composition is that, in classical statistical learning theory, memorization is typically viewed as harmful for generalization. For this reason, we find it especially interesting to identify and analyze a setting in which memorization can actually help compositional generalization.
>
> In our theoretical setup, the long-tail features are correctly specified and essentially independent, so memorizing more of these features cannot hurt composition by construction. Under these assumptions, spurious correlations are intentionally excluded from the model and therefore fall outside the scope of our guarantees.
>
> We agree that there are scenarios where memorization can hurt composition, particularly when the model memorizes spurious or mislabeled long-tail features. For example, if a rare feature is spuriously correlated with the label in the training distribution but not in the OOD regime, then memorizing this feature and composing it on new inputs will hurt OOD performance. In other words, the same mechanism that helps in our ''well-specified'' setting can become harmful when the long-tail features are misaligned with the true generative structure.
>
> In the revision, we have added a short discussion of this point.
>
> [1] Jiachen Zhao, In-Context Exemplars as Clues to Retrieving from Large Associative Memory, Neural Conversational AI @ ICML 2023.
>
> [2] Weimin Wu et al., In-Context Learning as Conditioned Associative Memory Retrieval, ICML 2025.

---

### Meta-Review · Area_Chair_hoHP · 2026-01-06

**Summary:**

This paper shows that memorization of rare examples can meaningfully support generalization, both theoretically and empirically. It extends the singleton memorization framework of Feldman et al. by linking memorization to one-shot and compositional generalization in overparameterized models. The experimental design—from linear settings to MNIST and Omniglot—clearly illustrates how architecture mediates the transition from memorization to composition. The theoretical analysis is sound and the presentation is clear and well motivated. While one reviewer raises concerns that appear to stem from a misunderstanding of the paper’s scope, the overall strength of the work supports acceptance as a poster.

**Reviewer Concerns:**

The main weakness concerns the limited experimental scope, with the reviewer suggesting that the authors further connect memorization-based composition to more realistic and complex tasks. There are also questions about when memorization might harm composition, especially in nonlinear settings and in the presence of spurious correlations. Additionally, the paper’s notion of memorization differs from influence-based definitions common in the literature, which prompted requests for clarification. That said, these concerns are largely about scope and framing, and the authors’ responses address them adequately without undermining the core technical contributions.

**Reviewer Scores:**

One reviewer rejected the paper. I believe that after reading the rebuttal, he or she will improve its score.

---

### Decision · Program_Chairs · 2026-01-26

Accept (Poster)